# Fuzzy Tiling Activations: A Simple Approach to Learning Sparse Representations Online

**Yangchen Pan**
University of Alberta
pan6@ualberta.ca

**Kirby Banman**
University of Alberta
kdbanman@ualberta.ca

**Martha White**
University of Alberta
whitem@ualberta.ca

## Abstract

Recent work has shown that sparse representations—where only a small percentage of units are active—can significantly reduce interference. Those works, however, relied on relatively complex regularization or meta-learning approaches, that have only been used offline in a pre-training phase. In this work, we pursue a direction that achieves sparsity by design, rather than by learning. Specifically, we design an activation function that produces sparse representations deterministically by construction, and so is more amenable to online training. The idea relies on the simple approach of binning, but overcomes the two key limitations of binning: zero gradients for the flat regions almost everywhere, and lost precision—reduced discrimination—due to coarse aggregation. We introduce a Fuzzy Tiling Activation (FTA) that provides non-negligible gradients and produces overlap between bins that improves discrimination. We first show that FTA is robust under covariate shift in a synthetic online supervised learning problem, where we can vary the level of correlation and drift. Then we move to the deep reinforcement learning setting and investigate both value-based and policy gradient algorithms that use neural networks with FTAs, in classic discrete control and Mujoco continuous control environments. We show that algorithms equipped with FTAs are able to learn a stable policy faster without needing target networks on most domains. [1]

## 1 Introduction

Representation learning in online learning systems can strongly impact learning efficiency, both positively due to generalization but also negatively due to interference (Liang et al., 2016; Heravi, 2019; Le et al., 2017; Liu et al., 2019; Chandak et al., 2019; Caselles-Dupré et al., 2018; Madjiheurem & Toni, 2019). Neural networks particularly suffer from interference—where updates for some inputs degrade accuracy for others—when training on temporally correlated data (McCloskey & Cohen, 1989; French, 1999; Kemker et al., 2018).

Recent work (Liu et al., 2019; Ghiassian et al., 2020; Javed & White, 2019; Rafati & Noelle, 2019; Hernandez-Garcia & Sutton, 2019), as well as older work (McCloskey & Cohen, 1989; French, 1991), have shown that sparse representation can reduce interference in training parameter updates. A sparse representation is one where only a small number of features are active, for each input (Cheng et al., 2013). Each update only impacts a small number of weights and so is less likely to interfere with many state values. Further, when constrained to learn sparse features, the feature vectors are more likely to be orthogonal (Cover, 1965), which further mitigates interference. The learned features can still be highly expressive, and even more interpretable, as only a small number of attributes are active for a given input.

However, learning sparse representations online remains relatively open. Some previous work has relied on representations pre-trained before learning, either with regularizers that encourage sparsity (Tibshirani, 1996; Xiang et al., 2011; Liu et al., 2019) or with meta-learning (Javed & White, 2019). Other work has trained the sparse-representation neural network online, by using sparsity regularizers online with replay buffers (Hernandez-Garcia & Sutton, 2019) or using a winner-take-all strategy where all but the top activations are set to zero (Rafati & Noelle, 2019). Hernandez-Garcia &

---

[1]Code is available at https://github.com/yannickycpan/reproduceRL.git

Sutton (2019) found that many of these sparsity regularizers were ineffective for obtaining sparse representations without high levels of dead neurons, though the regularizers did still often improve learning. The Winner-Take-All (WTA) approach is non-differentiable, and there are mixed results on it's efficacy, some positive (Rafati & Noelle, 2019) and some negative (Liu et al., 2019). Finally, kernel representations can be used online, and when combined with a WTA approach, provide sparse representations. There is some evidence that using only the closest prototypes—and setting kernel values to zero for the further prototypes—may not hurt approximation quality (Schlegel et al., 2017). However, kernel-based methods can be difficult to scale to large problems, due to computation and difficulties in finding a suitable distance metric. Providing a simpler approach to obtain sparse representations, that are easy to train online, would make it easier for researchers from the broad online learning community to adopt sparse representations and further explore their utility.

In this work, we pursue a strategy for what we call *natural sparsity*—an approach where we achieve sparsity by design rather than by encoding sparsity in the loss. We introduce an activation function that facilitates sparse representation learning in an end-to-end manner without the need of additional losses, pre-training or manual truncation. Specifically, we introduce a Fuzzy Tiling Activation (FTA) function that naturally produce sparse representation with controllable sparsity and can be conveniently used like other activation functions in a neural network. FTA relies on the idea of designing a differentiable approximate binning operation—where inputs are aggregated into intervals. We prove that the FTA guarantees sparsity by construction. We empirically investigate the properties of FTA in an online supervised learning problem, where we can carefully control the level of correlation. We then empirically show FTA's practical utility in a more challenging online learning setting—the deep Reinforcement Learning (RL) setting. On a variety of discrete and continuous control domains, deep RL algorithms using FTA can learn more quickly and stably compared to both those using ReLU activations and several online sparse representation learning approaches.

## 2 PROBLEM FORMULATION

FTA is a generic activation that can be applied in a variety of settings. A distinct property of FTA is that it does not need to learn to ensure sparsity; instead, it provides an immediate, deterministic sparsity guarantee. We hypothesize that this property is suitable for handling highly nonstationary data in an online learning setting, where there is highly correlated data stream and a strong need for interference reduction. We therefore explicitly formalize two motivating problems: the online supervised learning problem and the reinforcement learning (RL) problem.

**Online Supervised Learning problem setting**. The agent observes a temporally correlated stream of data, generated by a stochastic process $\{(X_t, Y_t)\}_{t \in \mathbb{N}}$, where the observations $X_t$ depend on the past $\{X_{t-i}\}_{i \in \mathbb{N}}$. In our supervised setting, $X_t$ depends only on $X_{t-1}$, and the target $Y_t$ depends only on $X_t$ according to a stationary underlying mean function $f(x) = \mathbb{E}[Y_t | X_t = x]$. On each time step, the agent observes $X_t$, makes a prediction $f_\theta(X_t)$ with its parameterized function $f_\theta$, receives target $Y_t$ and incurs a prediction error. The goal of the agent is to approximate function $f$—the ideal predictor—by learning from correlated data in an online manner, unlike standard supervised learning where data is independent and identically distributed (iid).

**RL problem setting**. We formalize the interaction using Markov decision processes (MDPs). An MDP consists of $(\mathcal{S}, \mathcal{A}, \mathbb{P}, R, \gamma)$, where $\mathcal{S}$ is the state space, $\mathcal{A}$ is the action space, $\mathbb{P}$ is the transition probability kernel, $R$ is the reward function, and $\gamma \in [0, 1]$ is the discount factor. At each time step $t = 1, 2, \ldots$, the agent observes a state $s_t \in \mathcal{S}$ and takes an action $a_t \in \mathcal{A}$. Then the environment transits to the next state according to the transition probability distribution, i.e., $s_{t+1} \sim \mathbb{P}(\cdot | s_t, a_t)$, and the agent receives a scalar reward $r_{t+1} \in \mathbb{R}$ according to the reward function $R : \mathcal{S} \times \mathcal{A} \times \mathcal{S} \to \mathbb{R}$. A policy is a mapping from a state to an action (distribution) $\pi : \mathcal{S} \times \mathcal{A} \to [0, 1]$. For a given state-action pair $(s, a)$, the action-value function under policy $\pi$ is defined as $Q_\pi(s, a) = \mathbb{E}[G_t | S_t = s, A_t = a; A_{t+1:\infty} \sim \pi]$ where $G_t \stackrel{\text{def}}{=} \sum_{t=0}^{\infty} \gamma^t R(s_t, a_t, s_{t+1})$ is the return of a sequence of transitions $s_0, a_0, s_1, a_1, \ldots$ by following the policy $\pi$.

The goal of an agent is to find an optimal policy that obtains maximal expected return from each state. The policy is either directly learned, as in policy gradient methods (Sutton et al., 1999; Sutton & Barto, 2018), or the action-values are learned and the policy inferred by acting greedily with respect to the action-values, as in Q-learning (Watkins & Dayan, 1992). In either setting, we often parameterize the policy/value function by a neural network (NN). For example, Deep $Q$ Networks (DQN) (Mnih

et al., 2015) parameterizes the action-value function $Q_\theta : \mathcal{S} \times \mathcal{A} \mapsto \mathbb{R}$ by a NN. The bootstrap target for updating a state-action value is computed by using a separate target NN $Q_{\theta^-} : \mathcal{S} \times \mathcal{A} \mapsto \mathbb{R}$ parameterized by $\theta^-$: $y_t = r_{t+1} + \gamma \max_{a'} Q_{\theta^-}(s_{t+1}, a')$. The target NN parameter $\theta^-$ is updated by copying from $\theta$ every certain number of time steps.

Online deep RL control problems can be highly nonstationary, for two primary reasons. First, the environment itself could be highly nonstationary, or alternatively, partially observable. Second, the data distribution is constantly shifting because of both the changing policy and shifting training targets. The latter can be mitigated by using a target NN as described above, which has become critical in successfully training many deep RL algorithms. However, it potentially slows learning as the new information is not immediately used to update action-values; instead, the slower moving and potentially out-dated target NN is used. Several works reported that successful training without a target NN can improve sample efficiency of RL algorithms (Liu et al., 2019; van Hasselt et al., 2018; Fan et al., 2020; Kim et al., 2019; Rafati & Noelle, 2019; Fan et al., 2020; Ghiassian et al., 2020). We show that deep RL algorithms using our activation is able to achieve superior performance without using a target NN, indicating the benefit of applying our method to nonstationary, online problems.

## 3 BINNING WITH NON-NEGLIGIBLE GRADIENTS

In this section, we develop the *Fuzzy Tiling Activation* (FTA), as a new modular component for neural networks that provides sparse representations. We first introduce a new way to compute the binning of an input, using indicator functions. This activation provides guaranteed sparsity but has a gradient of zero almost everywhere. Then, we provide a smoothed version, resulting in non-negligible gradients that make it compatible with back-propagation algorithms. We then prove that the fuzzy version is still guaranteed to provide sparse representation and the sparsity can be easily tuned.

### 3.1 TILING ACTIVATION

The tiling activation inputs a scalar $z$ and outputs a binned vector. This vector is one-hot, with a 1 in the bin corresponding to the value of $z$, and zeros elsewhere. Note that a standard activation typically maps a scalar to a scalar. However, the tiling activation maps a scalar to a vector, as depicted in Figure 1(a). This resembles tile coding, which inspires the name Tiling Activation; to see this connection, we include a brief review of tile coding in the Appendix A.1. In this section, we show how to write the tiling activation compactly, using element-wise max and indicator functions.

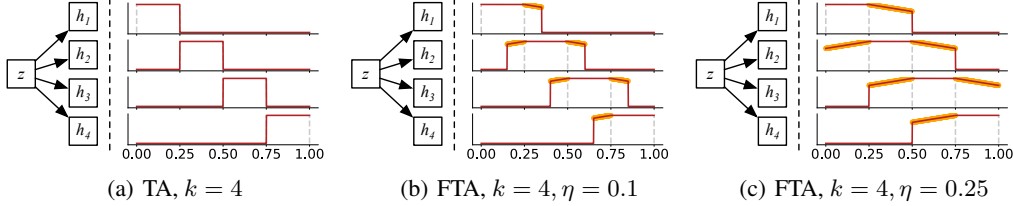

(a) TA, $k = 4$      (b) FTA, $k = 4, \eta = 0.1$      (c) FTA, $k = 4, \eta = 0.25$

Figure 1: a) The regular TA mapping $\mathbb{R} \to \mathbb{R}^k$, with each output element $h_i$ corresponds to a different bin. b) The FTA with $\eta > 0$, permitting both overlap in activation, and nonzero gradient between the vertical red and gray lines. c) Larger values for $\eta$ extends the sloped lines further from either side of each plateau, increasing the region that has non-negligible gradients.

Assume we are given a range $[l, u]$ for constants $l, u \in \mathbb{R}$, where we expect the input $z \in [l, u]$. The goal is to convert the input, to a one-hot encoding, with evenly spaced bins of size $\delta \in \mathbb{R}^+$. Without loss of generality, we assume that $u - l$ is evenly divisible by $\delta$; if it is not, the range $[l, u]$ could be slightly expanded, evenly on each side, to ensure divisibility. Define the $k$-dimensional tiling vector

$$\mathbf{c} \overset{\text{def}}{=} (l, l + \delta, l + 2\delta, ..., u - 2\delta, u - \delta). \tag{1}$$

where $k = (u - l)/\delta$. The **tiling activation** is defined as

$$\phi(z) \overset{\text{def}}{=} 1 - I_+(\max(\mathbf{c} - z, 0) + \max(z - \delta - \mathbf{c}, 0)) \tag{2}$$

where $I_+(\cdot)$ is an indicator function, which returns 1 if the input is positive, and zero otherwise. The indicator function for vectors is applied element-wise. In Proposition 1, we prove that $\phi$ returns a

binned encoding: if $c_i < z < c_{i+1}$, then $\phi(z)$ returns $e_i$ the one-hot (standard basis) vector with a 1 in the $i$-th entry and zero elsewhere. For values of $z$ that fall on the boundary, $z = c_i$, the encoding returns a vector with ones in both the $i-1$th and $i$th entries. Consider the below example for intuition.

**Example**. Assume $[l, u] = [0, 1]$ and set the tile width to $\delta = 0.25$. Then the tiling vector $\mathbf{c}$ has four tiles ($k = 4$): $\mathbf{c} = (0, 0.25, 0.5, 0.75)$. If we apply the tiling activation to $z = 0.3$, because $0.25 < 0.3 < 0.5$, the output should be $(0, 1, 0, 0)$. To see $\phi(z)$ does in fact return this vector, we compute each max term

$$\max(\mathbf{c} - z, 0) = (0, 0, 0.2, 0.45) \quad \text{and} \quad \max(z - \delta - \mathbf{c}, 0) = \max(0.05 - \mathbf{c}, 0) = (0.05, 0, 0, 0).$$

The addition of the two is $(0.05, 0, 0.2, 0.45)$ and so $1 - I_+(0.05, 0, 0.2, 0.45) = 1 - (1, 0, 1, 1) = (0, 1, 0, 0)$. The first max extracts those components in $\mathbf{c}$ that are strictly greater than $z$, and the second max extracts those strictly less than $z$. The addition gives the bins that are strictly greater and strictly less than the bin for $z$, leaving only the entry corresponding to that activated bin as 0, with all others positive. The indicator function sets all nonzero entries to one and then using one minus this indicator function's output provides us the desired binary encoding. We rigorously characterize the possible output cases for the activation in the Appendix A.2.1.

### 3.2 FUZZY TILING ACTIVATION (FTA)

The Tiling Activation provides a way to obtain sparse, binary encodings for features learned within a NN. Unfortunately, the tiling activation has a zero derivative almost everywhere as visualized in Figure 1(a). In this section, we provide a fuzzy tiling activation, that has non-zero derivatives and so is amenable to use with backpropagation.

To design the FTA, we define the fuzzy indicator function[2]

$$I_{\eta,+}(x) \stackrel{\text{def}}{=} I_+(\eta - x)x + I_+(x - \eta) \tag{3}$$

where $\eta$ is a small constant for controlling the sparsity. The first term $I_+(\eta - x)$ is 1 if $x < \eta$, and 0 otherwise. The second term $I_+(x - \eta)$ is 1 if $x > \eta$, and 0 otherwise. If $x < \eta$, then $I_{\eta,+}(x) = x$, and else $I_{\eta,+}(x) = 1$. The original indicator function $I_+$ can be acquired by setting $\eta = 0$. When $\eta > 0$, the derivative is non-zero for $x < \eta$, and zero otherwise. Hence the derivative can be propagated backwards through those nonzero entries. Using this fuzzy indicator function, we define the following **Fuzzy Tiling Activation** (FTA)

$$\phi_\eta(z) \stackrel{\text{def}}{=} 1 - I_{\eta,+}(\max(\mathbf{c} - z, 0) + \max(z - \delta - \mathbf{c}, 0)) \tag{4}$$

where again $I_{\eta,+}$ is applied elementwise.

We depict FTA with different $\eta$s in Figure 3.1. For the smaller $\eta$, the FTA extends the activation to the neighbouring bins. The activation in these neighbouring bins is sloped, resulting in non-zero derivatives. For this smaller $\eta$, however, there are still regions where the derivative is zero (e.g., $z = 0.3$ in Figure 1(b)). The regions where derivatives are non-zero can be expanded by increasing $\eta$ as shown in Figure 1(c). Hence we can adjust $\eta$ to control the sparsity level as we demonstrate in Section A.5.

Figure 2 shows a neural network with FTA applied to the second hidden layer and its output $y$ is linear in the sparse representation. FTA itself does not introduce any new training parameters, just like other activation functions. For input $\mathbf{x}$, after computing first layer $\mathbf{h}_1 = \mathbf{x}\mathbf{W}_1$, we apply $\phi_\eta(\mathbf{z})$ to $\mathbf{h}_1\mathbf{W}_2 \in \mathbb{R}^d$ to get the layer $\mathbf{h}_2$ of size $kd$. This layer consists of stacking the $k$-dimensional sparse encodings, for each element in $\mathbf{h}_1\mathbf{W}_2$.

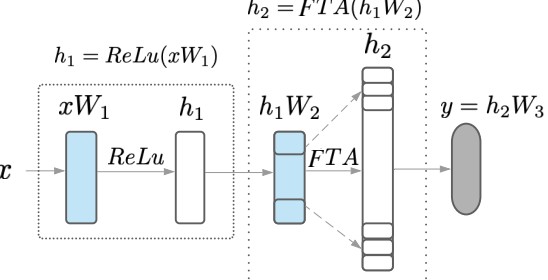

Figure 2: A visualization of an FTA layer

### 3.3 GUARANTEED SPARSITY FROM THE FTA

We now show that the FTA maintains one of the key properties of the tiling activation: sparsity. The distinction with many existing approaches is that our sparsity is guaranteed by design and hence is

---

[2]The word fuzzy reflects that an input can partially activate a tile, with a lower activation than 1, as an analogy to the concept of partial inclusion and degrees of membership from fuzzy sets.

not a probabilistic guarantee. We first characterize the vectors produced by the FTA, in Proposition 2 with proof in Appendix A.2.2. Then, we provide an upper bound on the proportion of nonzero entries in the generated vector in Theorem 1, with in Appendix A.2.3.

**Assumption 1.** $\delta < u - l$, where $k\delta = u - l$ for $k \in \mathbb{N}$.

**Theorem 1** (Sparsity guarantee for FTA.). *For any $z \in [l, u], \eta > 0$, $\phi_\eta(z)$ outputs a vector whose number of nonzero entries $\|\phi_\eta(z)\|_0$ satisfies:*

$$\|\phi_\eta(z)\|_0 \leq 2 \left\lfloor \frac{\eta}{\delta} \right\rfloor + 3$$

**Corollary 1.** *Let $\rho \in [0, 1)$ be the desired sparsity level: the maximum proportion of nonzero entries of $\phi_\eta(z), \forall z \in [l, u]$. Assume $\rho k \geq 3$, i.e., some inputs have three active indices or more (even with $\eta = 0$, this minimal active number is 2). Then $\eta$ should be chosen such that*

$$\left\lfloor \frac{\eta}{\delta} \right\rfloor \leq \frac{k\rho - 3}{2} \quad \text{or equivalently} \quad \eta \leq \frac{\delta}{2} \left( \lfloor k\rho \rfloor - 1 \right) \tag{5}$$

As an example, for $k = 100$, $\delta = 0.05$ and a desired sparsity of at least 10% ($\rho = 0.1$), we can use $\eta = \frac{0.05}{2}(\lfloor 100 \times 0.1 \rfloor - 1) = 0.225$. Note that the bound in Theorem 1 is loose in practice as the bound is for any input $z$. In Appendix A.2.3 and A.5, we theoretically and empirically show that the actual sparsity is usually lower than the upper bound, and quite consistent across inputs.

## 4 EXPERIMENTS IN SUPERVISED LEARNING UNDER COVARIATE SHIFT

In this section, we focus on testing the hypothesis that FTA provides representations that are more robust to learning online on correlated data. Specifically, we hypothesize that convergence speed and stability for ReLU networks suffer under strongly correlated training data, whereas comparable FTA networks are nearly unaffected. We create a synthetic supervised problem with a relatively simply target function, and focus the investigation on the impact of a drifting distribution on inputs, which results both in covariate shift and creates temporal correlation during training. We also report results on two benchmark image classification tasks in the Appendix A.6.

The Piecewise Random Walk Problem has Gaussian $X_t \sim \mathcal{N}(S_t, \beta^2)$ with fixed variance $\beta^2$ and a mean $S_t$ that drifts every $T$ steps. More precisely, the mean $S_t$ stays fixed for $T$ timesteps, then takes a step according to a first order autoregressive random walk: $S_t = (1 - c)S_t + Z_t$ where $c \in (0, 1]$ and $Z_t \sim \mathcal{N}(0, \sigma^2)$ for fixed variance $\sigma^2$. If $c = 0$, then this process is a standard random walk; otherwise, with $c < 1$, it keeps $S_t$ in a bounded range with high probability. For $x_t \sim X_t$, the training label $y_t$ is defined as $y_t = \sin(2\pi x_t^2)$.

This process is designed to modulate the level of correlation—which we call *correlation difficulty*— without changing the equilibrium distribution over $X_t$. As the correlation difficulty $d$ varies from 0 to 1, the training data varies from low to high correlation: $d = 0$ recovers iid sampling. All $d \in [0, 1)$ share the same equilibrium distribution in $X_t$. $X_t$ is ergodic and has Gaussian equilibrium distribution $\mathcal{N}(0, \xi^2)$, with variance $\xi^2$ dependent upon $\beta^2, \sigma^2$ and $c$. In particular, the visitation distribution $X_t$ for any training run will converge to the equilibrium distribution. This ensures that measuring loss with respect to the stationary distribution is a fair comparison, because the visitation distribution of $X_t$ is identical across all settings. We depict sample trajectories with low $d$ and high $d$ in Figure 3(a). For a rigorous construction of $X_t$ and $S_t$, and justification for this equilibrium distribution and implementation details, see Appendix A.7.

We measure the mean squared error over the equilibrium distribution in $X_t$, for neural networks using FTA and ReLU activations across a range of correlation difficulty values. In Figure 3, we can see that FTA outperforms ReLU in two ways. First, FTA converges to a lower loss with less variance across all correlation difficulties. Second, FTA only marginally suffers under high difficulties $d > 0.9$, whereas the performance of ReLU begins to deteriorate for relatively mild $d > 0.5$.

Note that the FTA reaches a lower error, even on iid data ($d = 0$). We hypothesize this gap arises because the networks are trained online, with one sample from each $X_t$ being used for each weight update. Figure 23 in the Appendix supports this hypothesis, with the gap vanishing in an identical experiment where 50 samples are drawn from each $X_t$.

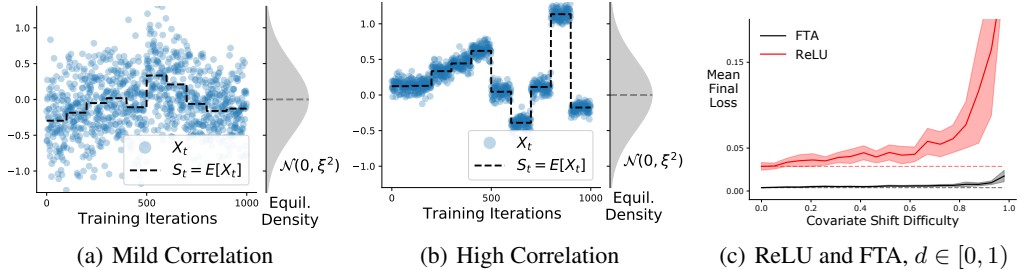

(a) Mild Correlation     (b) High Correlation     (c) ReLU and FTA, $d \in [0, 1)$

Figure 3: (a) and (b) contain sample trajectories of $X_t$ which are (a) mildly correlated with $d = 0.41$ and (b) severely correlated with $d = 0.98$. Both share the same equilibrium distribution (in gray). (c) A plot of the prediction error (under the equilibrium distribution), averaged over the final 2.5K iterations, across a range of difficulty settings. All networks are trained for 20k online updates. The lines correspond to the mean of 30 runs, with the shaded region corresponding to 99.9% confidence intervals. The iid setting $d = 0$ is shown as a dotted line for baseline comparison.

## 5   EXPERIMENTAL RESULTS IN REINFORCEMENT LEARNING

In this section, we empirically study the effect of using FTA in RL. First, we show overall performance on several benchmark discrete and continuous control environments. Second, we compare our method with other simple strategies to obtain sparse representations. Third, we provide insight into different hyper-parameter choices of FTA and suggest potential future directions. Appendix A.4 includes details for reproducing experiments and Appendix A.5 has additional RL experiments.

### 5.1   ALGORITHMS AND NAMING CONVENTIONS

All the algorithms use a two-layer neural network, with the primary difference being the activation used on the last hidden layer. See Appendix A.5 for results using FTA in all the hidden layers. DQN is used for the discrete action environments, and Deep Deterministic Policy Gradient (DDPG) (Lillicrap et al., 2016) for continuous action environments. On each step, all algorithms sample a mini-batch size of 64 from an experience replay (Lin, 1992; Mnih et al., 2015) buffer with maximum size 100k. Note that we keep the same FTA setting across *all* experiments: we set $[l, u] = [-20, 20]$, $\delta = \eta = 2.0$, and hence $\mathbf{c} = \{-20, -18, -16, ..., 18\}$, $k = 40/2 = 20$.

We first compare to standard DQN agents, with the same architectures except the last layer.
**DQN**: DQN with tanh or ReLU on the last layer (best performance reported).
**DQN-FTA**: DQN with FTA on the last layer.
**DQN-Large**: DQN, but with the last layer of the same size as DQN-FTA. If DQN has a last (i.e., the second) hidden layer of size $d$, then DQN-FTA has a last layer of size $dk$, since the FTA activation simply expands the number of features due to binning. Hence, we include DQN-Large with the same feature dimension as DQN-FTA in the last hidden layer. Note that DQN-Large has $k$ times more parameters in this last hidden layer than DQN or DQN-FTA. [3]

We also compare to several simple strategies to obtain local or sparse features. Radial basis functions (RBFs) have traditionally been used to obtain local features in RL, and recent work has used $\ell_2$ and $\ell_1$ regularization directly on activations as a simple baseline (Arpit et al., 2016; Liu et al., 2019). All of these strategies have the same sized last layer as the sparse feature dimension of DQN-FTA.
**DQN-RBF**: DQN using radial basis functions (RBFs) on the last layer, with the centers defined by the same $\mathbf{c}$ as FTA: $\phi(z) = [\exp\left(-\frac{(z - \mathbf{c}_1)^2}{\sigma}\right), \dots, \exp\left(-\frac{(z - \mathbf{c}_k)^2}{\sigma}\right)]$ where $\sigma$ is the bandwidth parameter.
**DQN-L2/L1**: DQN with $\ell_2$ or $\ell_1$ regularization on the activation functions of the final hidden layer where there is the same number of units as that in DQN-Large.

To the best of our knowledge, no suitable sparse representation approaches exist for RL. SR-NNs for RL were only developed for offline training (Liu et al., 2019). That work also showed that k-sparse NNs (Makhzani & Frey, 2013) and Winner-Take-All NNs (Makhzani & Frey, 2015) performed significantly worse than $\ell_2$ regularization, where $\ell_2$ actually performed quite well in most of their

---

[3]Please refer to Figure 2. FTA does not augment the number of training parameters in the hidden layer where it is applied; it only augments the training parameters in the immediately next layer.

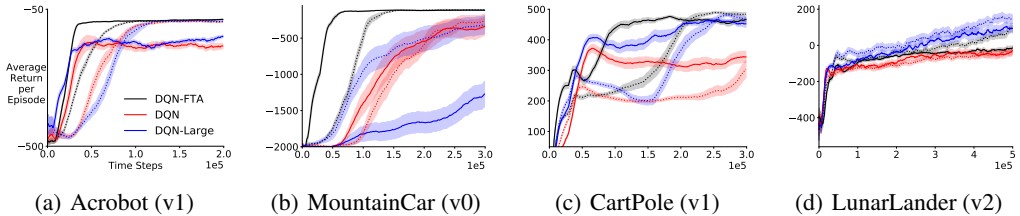

| (a) Acrobot (v1) | (b) MountainCar (v0) | (c) CartPole (v1) | (d) LunarLander (v2) |

Figure 4: Evaluation learning curves of **DQN-FTA(black)**, **DQN(red)**, and **DQN-Large(blue)**, showing episodic return versus environment time steps. The **dotted** line indicates algorithms trained *with* target networks. The results are averaged over 20 runs and the shading indicates standard error. The learning curve is smoothed over a window of size 10 before averaging across runs.

experiments. One other possible option is to use Tile Coding NNs (Ghiassian et al., 2020), which first tile code inputs before feeding them into the neural network. This approach focuses on using discretization to break, or overcome, incorrect generalization in the inputs; their goal is not to learn sparse representations. This approach is complementary, in that it could be added to all the agents in this work. Nonetheless, because it is one of the only papers with a lightweight approach to mitigate interference in online RL, we do compare to it in the Appendix A.5.2.

## 5.2 OVERALL PERFORMANCE

In this section, we demonstrate the overall performance of using FTAs on both discrete and continuous control environments. Our goals are to investigate if we can 1) obtain improved performance with FTA, with fixed parameter choices across different domains; 2) improve stability in learning with FTA; and 3) see if we can remove the need to use target networks with FTA, including determining that learning can in fact be faster without target networks and so that it is beneficial to use FTA without them. All experiments are averaged over 20 runs (20 different random seeds), with offline evaluation performed on the policy every 1000 training/environment time steps.

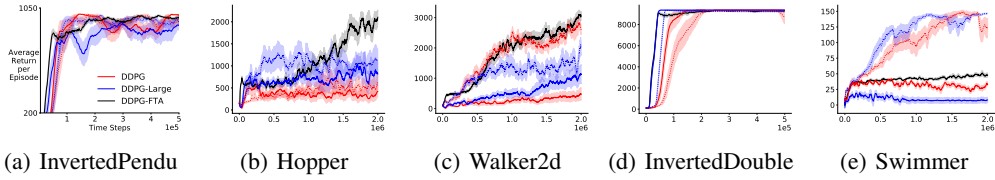

| (a) InvertedPendu | (b) Hopper | (c) Walker2d | (d) InvertedDouble | (e) Swimmer |

Figure 5: Evaluation learning curves of **DDPG-FTA(black)**, **DDPG(red)**, and **DDPG-Large(blue)**, averaged over 10 runs with shading indicating standard error. The **dotted** line indicates algorithms trained *with* target networks. The learning curve is smoothed over a window of size 30 before averaging across runs.

**Discrete control.** We compare performance on four discrete-action environments from Ope-nAI (Brockman et al., 2016): MountainCar, CartPole, Acrobot and LunarLander. We use $64 \times 64$ ReLU hidden units on all these domains. Since FTA has $k = 20$, this yields $64 \times 20 = 1280$ dimensional sparse features. Hence, DQN-Large, use two layers ReLU with size $64 \times 1280$.

We plot evaluation learning curves, averaged over 20 runs, in Figure 4. The results show the following. 1) With or without using a target network, DQN with FTA can significantly outperform the version without using FTA. 2) FTA has significantly lower variability across runs (smaller standard errors) in most of the figures. 3) DQN-FTA trained *without* a target network outperforms DQN-FTA trained *with* a target network, which indicates a potential gain by removing the target network. 4) Without using FTA, DQN trained without a target network cannot perform well in general, providing further evidence for the utility of sparse feature highlighted in previous works (Liu et al., 2019; Rafati & Noelle, 2019). 5) Simply using a larger neural network does not obtain the same performance improvements, and in some cases significantly degrades performance. The exception to these conclusions is LunarLander, where DQN-FTA performed similarly to DQN and actually performed slightly better with target networks. On deeper investigation, we found that this was due to using a tiling bound of $[-20, 20]$, which we discuss further in Section 5.4. When using $[-1, 1]$, DQN-FTA performs much better and results are consistent with the other domains.

**Continuous control.** We compare performance on continuous control tasks from Mujoco (Todorov et al., 2012). For these experiments, we use DDPG with exactly the same FTA settings as in the above discrete control domains to show its generality. Corresponding to the discrete control domains, we apply FTA to the critic network and do not use a target network to train it. As seen in Figure 5, on most domains, DDPG-FTA achieves comparable and sometimes significantly better performance to DDPG with a target network. However, Figure 5 (e) highlights that FTA is not always sufficient on its own to achieve superior performance. Factors such as the exploration strategy, other hyper-parameters, and neural network architecture may also play an important role on such challenging tasks.

## 5.3    COMPARISON WITH OTHER REPRESENTATION LEARNING APPROACHES

FTA provides clear benefits, but it is natural to ask if other simple strategies that provide sparse or local features could have provided similar benefits. We compare to DQN-RBF, DQN-L2 and DQN-L1, on the discrete action environments, shown in Figure 6. We find the following. 1) FTA performs consistently well across all environments using a fixed parameter setting; none of the other approaches achieve consistent performance, even though we tuned their parameters per environment. 2) Both the $\ell_1$ and $\ell_2$ approaches have a high variance across different random seeds. 3) The RBF variant can do better than the $\ell_1$ and $\ell_2$ approaches but is worse than our algorithm. It is a known issue that RBF is sensitive to the bandwidth parameter choice and we observe similar phenomenon. It is also known that the exponential activation can be problematic in the back-propagation process. We empirically investigate the overlap and instance sparsities of each algorithm in the Appendix A.5.

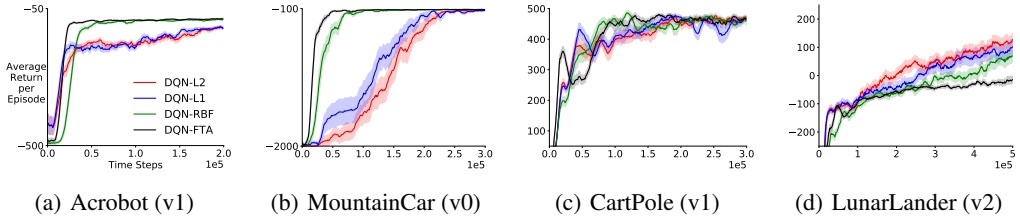

(a) Acrobot (v1)          (b) MountainCar (v0)          (c) CartPole (v1)          (d) LunarLander (v2)

Figure 6:    Evaluation learning curves of **DQN-FTA(black)**, **DQN-RBF(forest green)**, **DQN-L2(red)**, and **DQN-L1(blue)**, averaging over 20 runs with the shade indicating standard error. All algorithms are trained without using target networks. The learning curve is smoothed over a window of size 10 before averaging across runs.

## 5.4    CHOOSING THE HYPER-PARAMETERS FOR FTA

In this section, we provide some insight into selecting the hyper-parameters of FTA. We argue that $\eta = \delta$ can be used as a rule of thumb. Then we show that given an appropriate tiling bound (i.e., $u$ value), DQN-FTA performs well, with a reasonably large number of bins, i.e., a reasonably small tile width $\delta$. However, we do observe a certain level of sensitivity to the tiling bound, which may partially explain the worse performance of DQN-FTA on LunarLander and some Mujoco domains.

**Sparsity control parameter**. The purpose of the parameter $\eta$ is to provide a nonzero gradient for training an NN via backpropagation. Since FTA is an one-to-many mapping, it gives a nonzero gradient as long as any single element in the resulting vector provides a nonzero gradient. Setting $\eta = \delta$ (i.e., the tile width) is the minimum value to guarantee nonzero gradient as we visualize in the Figure 1(c) as long as FTA's input is within the tiling bound. In all of our experiments, we fix $\eta = \delta$ unless otherwise specified. In the Appendix A.5.4, we show that DQN-FTA does reasonably well across a broad range of $\eta, \delta$ parameter settings.

**Number of tiles/tile width**. In Figure 7(c), we show that on LunarLander, with $u = 1.0$, DQN-FTA performs well when the tile width $\delta = 2u/k$ is reasonably small (i.e., number of tiles $k$ is reasonably large). We show the learning curves of DQN-FTA with $u = 1.0$, in Figure 7(a), by using number of tiles from $k \in \{2, 4, 6, 8, 10, 12, 14, 16\}$, providing sparse feature dimension $64 \times k \in \{128, 256, 384, 512, 640, 768, 896, 1024\}$. We also examine the performance of DQN trained by using NN size $64 \times k$, for $k \in \{2, 4, 6, 8, 10, 12, 14, 16\}$. In Figure 7(b), one can see

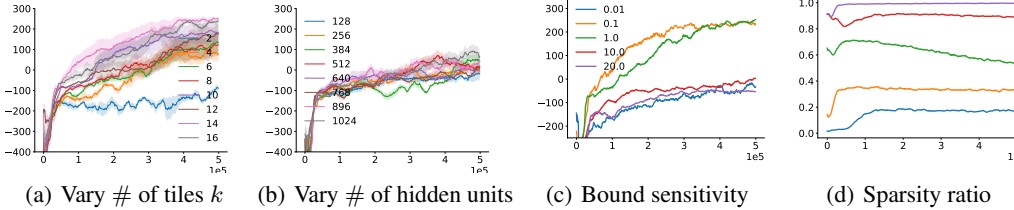

(a) Vary # of tiles $k$    (b) Vary # of hidden units    (c) Bound sensitivity    (d) Sparsity ratio

Figure 7: (a) Evaluation learning curves showing episodic return versus environment time steps of DQN-FTA using different number of tiles $k$ as labeled. This is equivalent to varying tile width $\delta$ as $\delta = 2u/k$. The results are averaged over 5 random seeds. (b) Evaluation learning curves of DQN without using a target NN as we change the number of the second hidden layer units as labeled in the figure. (c) Evaluation learning curves of DQN-FTA uses $[-u, u]$ as tiling bound where $u \in \{0.01, 0.1, 1.0, 10.0, 20.0\}$. The results are averaged over 10 runs. The standard error is not shown but is sufficient small to differentiate the two groups (i.e., $\{0.1, 1.0\}$ and $\{0.01, 10.0, 20.0\}$) corresponding to appropriate bound and too large/small bound. To generate this figure, we fix on using FTA with 20 tiles (i.e., tile width $\delta = 2u/20$). (d) The overlap-instance sparsity ratio v.s. environment time steps. The standard error is very small and is ignored. The results are averaged over 10 runs.

that increasing NN size does not provide a clear benefit to DQN; however, DQN-FTA performs significantly better as $k$ increases, in Figure 7(a).

**Sensitivity to tiling bound**. We find that the tiling bound $[l, u]$ can be sensitive. Intuitively, if we set $l, u$ extremely small, then the input of FTA may go out of the boundary and FTA provides zero gradient again. When we set the bound too large, many inputs may hit the same bins, both resulting in many dead neurons and increasing interference. In Figure 7(c), we see that very small $u = 0.01$ and big $u = 10, 20$ perform poorly in LunarLander, and interim values of $u = 0.1, 1$ perform well.

We further examine the corresponding representation interference measured by the overlap sparsity divided by instance sparsity, in Figure 7(d). Instance sparsity is the proportion of nonzero entries in the feature vector for each instance. Overlap sparsity (French, 1991) is defined as $overlap(\phi, \phi') = \sum_i I(\phi_i \neq 0) I(\phi'_i \neq 0)/(kd)$, given two $kd$-dimensional sparse vectors $\phi, \phi'$. Low overlap sparsity potentially indicates less feature interference between different input samples. Overlap sparsity divided by instance sparsity represents the proportion of overlapped entries among activated ones. We can see in Figure 7(d) that large $u$ increases this ratio, indicating increased interference, which may explain the worse performance of DQN-FTA with $u = 10, 20$ in Figure 7(c). On the other extreme, when the bound is too small, $u = 0.01$, the sparsity ratio is low but performance is poor. This is likely because generalization actually becomes too low at this level, reducing sample efficiency. We refer to Appendix A.5.6 for additional experiments about gradient interference.

## 6 DISCUSSION

In this work, we proposed the idea of natural sparsity, aiming at achieving sparsity without learning in a deep learning setting. We design an activation by drawing on the idea of binning which produces a one-hot encoding of an input. We provide a Fuzzy Tiling Activation (FTA) with a sparsity control mechanism that enables backpropagation of gradients, and potentially improves generalization by increasing active feature units. We show that the FTA still has sparsity guarantees, related to the choice of $\eta$. The FTA provides sparse representations by construction, and so it is much easier to use when learning online than conventional sparse representation learning methods. We highlight that FTA is robust to high levels of covariate shift, in a synthetic supervised learning problem. We then show across several discrete and continuous control RL environments that using the FTA significantly improve learning efficiency and stability, and in most cases even removes the need of target networks.

Our work suggests several promising future research directions. The first is to further investigate the choices for the tiling bound. An adaptive approach for the tiling bound is particularly relevant for ease-of-use, as we found more sensitivity to this choice. Second, FTA is actually complementary to approaches (Liu et al., 2018; Riemer et al., 2019; Javed & White, 2019) designed to explicitly mitigate interference. An interesting direction is to combine FTA with these methods to reduce interference. Last, it is important to study how we should balance generalization and discrimination. We observe that an algorithm's performance degrades when either the overlap sparsity is too high or too low. Learning sparse representation with an appropriate overlap sparsity may be most desirable.

## 7 ACKNOWLEDGEMENTS

We would like to thank all anonymous reviewers for their helpful feedback. We thank Han Wang for carefully reading our paper and providing helpful suggestions to improve our presentation. We acknowledge the funding from the Canada CIFAR AI Chairs program and Alberta Machine Intelligence Institute.

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

## A  APPENDIX

This appendix includes the following contents:

1. Section A.1 briefly reviews tile coding which inspires FTA and the naming.

2. Section A.2 shows the proofs for theorems about sparsity guarantee in this paper.

3. Section A.3 discusses an alternative way to handle the case when the inputs of FTA go out of the boundary of the tiling vector $\mathbf{c}$.

4. Section A.4 includes experimental details of Section 5 for reproducible research.

5. Section A.5 presents additional experiments in reinforcement learning setting.

6. Section A.6 reports the results of FTA on two popular image classification datasets: Mnist (LeCun & Cortes, 2010) and Mnistfashion (Xiao et al., 2017).

7. Section A.7 includes the details of the synthetic supervised learning experiment from Section 4.

### A.1  TILE CODING REVIEW

We give a brief review of tile coding here, as tile coding inspires our Fuzzy Tiling Activation (and its naming).

Tile coding is a generalization of state aggregation, that uses multiple tilings (aggregations) to improve discrimination. For input $z \in [0, 1]$, state-aggregation would map $z$ to a one-hot vector with a 1 in the corresponding bin (bin can be also called tile), with $k$ the number of bins discretized into intervals of length $\delta$. In tile coding, multiple such tilings of the input are concatenated, where each tiling is offset by a small amount. This is depicted in Figure 8, where we show two tilings, one covering from $[-0.05, 1]$ and the other from $[0.0, 1.05]$ both with $k = 4$ and $\delta = 1.05/4 = 0.2625$. The resulting feature vector is a concatenation of these two tilings, to produce 8 features. Two nearby inputs $z = 0.3$ and $z' = 0.55$ would be aggregated together in state-aggregation, making it impossible to produce different values for those inputs. Tile coding, on the other hand, would cause them to share one feature in the second tiling, but each have a distinct feature that enables them to have different values. The large bins/tiles allows for some information sharing, for fast generalization; and the overlapping tilings allows for improved discrimination.

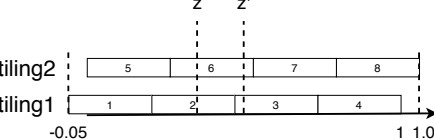

Figure 8:  Tile coding maps a scalar to an 8 dimensional binary vector. For example, $z$ activates the second tile on both tilings which gives the feature vector $(0, 1, 0, 0, 0, 1, 0, 0)$. Similarly, $z' \mapsto (0, 0, 1, 0, 0, 1, 0, 0)$.

Unfortunately, tile coding can scale exponentially in the dimension $d$ of the inputs. Each dimension is discretized to a granularity of $\delta$ ($k$ regions), with the cross-product between all these regions resulting in $k^d$ bins. One natural approach to make binning more scalable and more practically usable, is to combine it with neural networks (NN). Our activation function—Fuzzy Tiling Activation—enables a differentiable binning operation.

It is known that those simple domains which are able to use tile coding never require a separate slowly moving weight vector (i.e., the counterpart of a target network in linear function approximation setting) (Sutton, 1996; Sutton & Barto, 2018). Based on this observation, Liu et al. (2019) indicates that in a deep learning setting, when using sparse representation, the target network can be removed and the sample efficiency can be improved, coinciding with many of our empirical results. It is sensible that the target network possibly slower down learning. Because at each environment time step, the new information is not immediately utilized to estimate the bootstrap target and this could slower down learning. Accurately estimating bootstrap targets may be highly beneficial in Dyna-style model-based reinforcement learning Sutton (1991); Sutton et al. (2008), as the planning stage typically involves efficient improvement of value estimates (Moore & Atkeson, 1993; Pan et al., 2018; Gu et al., 2016; Pan et al., 2019). It may be an interesting future direction to carefully study the effect of removing the target network in Dyna-style planning.

## A.2 PROOFS

We now provide the proof for Proposition 1, Proposition 2, Theorem 1 and Corollary 1 below.

### A.2.1 PROOF FOR PROPOSITION 1

For convenience, we define the $k$-dimensional one-hot vector $\mathbf{e}_i$ whose $i$th entry is one and otherwise zero.

**Proposition 1.** *Under Assumption 1, for any $z \in [l, u]$,*

1. *If $\mathbf{c}_i < z < \mathbf{c}_{i+1}$ for $i \in [k-1]$ **or** $c_k < z$, then $\phi(z) = \mathbf{e}_i$*

2. *If $z = \mathbf{c}_i$ for $i \in \{2, ..., k\}$, then $\phi(z) = \mathbf{e}_{i-1} + \mathbf{e}_i$*

3. *If $z = \mathbf{c}_1$, then $\phi(z) = \mathbf{e}_1$*

*Proof.* In all thee cases, the first max operation in $\phi$ is

$$\max(\mathbf{c} - z, 0) = (0, 0, ..., \mathbf{c}_{i+1} - x, \mathbf{c}_{i+2} - z, ..., \mathbf{c}_k - z).$$

To understand the output of the second max operation, we look at the three cases separately.
**Case 1:** Because $\mathbf{c}_i < z < \mathbf{c}_{i+1}$, we know that $\mathbf{c}_{i-1} < z - \delta < \mathbf{c}_i$. This implies the second max operation in $\phi$ is:

$$\max(z - \delta - \mathbf{c}, 0)$$
$$= (z - \delta - \mathbf{c}_1, z - \delta - \mathbf{c}_2, ..., z - \delta - \mathbf{c}_{i-1}, 0, 0, ..., 0)$$

Therefore, the sum of both max operations $\max(\mathbf{c} - z, 0) + \max(z - \delta - \mathbf{c}, 0)$ has positive elements everywhere except the $i$th position, which is zero. Hence $I_+(\max(\mathbf{c} - z, 0) + \max(z - \delta - \mathbf{c}, 0))$ gives a vector where every entry is 1 except the $i$th entry which is 0. Then $1 - I_+(\max(\mathbf{c} - z, 0) + \max(z - \delta - \mathbf{c}, 0)) = \mathbf{e}_i$.
**Case 2:** If $z = \mathbf{c}_i, i \in \{2, ..., k\}$, then $\mathbf{c}_{i-1} = z - \delta$, and

$$\max(z - \delta - \mathbf{c}, 0)$$
$$= (z - \delta - \mathbf{c}_1, z - \delta - \mathbf{c}_2, ..., z - \delta - \mathbf{c}_{i-2}, 0, ..., 0).$$

It follows that $\max(\mathbf{c} - z, 0) + \max(z - \delta - \mathbf{c}, 0)$ has exactly two zero entries, at indices $i - 1$ and $i$. This gives $1 - I_+(\max(\mathbf{c} - z, 0) + \max(z - \delta - \mathbf{c}, 0)) = \mathbf{e}_{i-1} + \mathbf{e}_i$, a vector with ones at indices $i - 1$ and $i$.
**Case 3:** When $z = \mathbf{c}_1$, $\max(z - \delta - \mathbf{c}, 0)$ is a zero vector and $\max(\mathbf{c} - z, 0)$ is positive everywhere except the first entry, which is zero. Again this gives $1 - I_+(\max(\mathbf{c} - z, 0) + \max(z - \delta - \mathbf{c}, 0)) = \mathbf{e}_1$. $\qquad\square$

### A.2.2 PROOF FOR PROPOSITION 2

**Proposition 2.** *Let $\mathcal{I}$ be the set of indices where $\phi(z)$ is active (from Theorem 1 we know it can only contain one or two indices). Under Assumption 1, for any $z \in [l, u]$, the function $\phi_\eta(z) = \phi(z) + \Delta$, where $\Delta$ is mostly zero, with a few non-zero entries. The vector $\Delta$ has the following properties:*

1. *$\Delta$ is zero at indices $i \in \mathcal{I}$, i.e., $\phi_\eta(z)$ equals $\phi(z)$ at indices $i \in \mathcal{I}$.*

2. *$\Delta$ is non-zero at indices $\{j | j \notin \mathcal{I}, j \in [k], 0 < z - \delta - \mathbf{c}_j \leq \eta, 0 < \mathbf{c}_j - z \leq \eta\}$.*

*Proof.* **Part 1.** Let the $i$th entry be active in $\phi(z)$. We have one of the three cases hold as stated in Theorem 1. Assume $\mathbf{c}_i < z < \mathbf{c}_{i+1}$. Note that

$$\max(\mathbf{c} - z, 0) = (0, 0, ..., \mathbf{c}_{i+1} - z, ..., \mathbf{c}_k - z)$$
$$\max(z - \delta - \mathbf{c}, 0) = (z - \delta - \mathbf{c}_1, ..., z - \delta - \mathbf{c}_{i-1}, 0, ..., 0),$$

taking the sum of the above two equations gives us a vector as following:

$$(z - \delta - \mathbf{c}_1, ..., z - \delta - \mathbf{c}_{i-1}, 0, \mathbf{c}_{i+1} - z, ..., \mathbf{c}_k - z) \tag{6}$$

Then applying $I_{\eta,+}(\cdot)$ to vector equation 6 gives us a vector retaining elements $\leq \eta$ and all other elements become 1. Hence the $i$th position is zero after applying $I_{\eta,+}(\cdot)$ to vector equation 6. Using

one minus this vector would give us a vector with only the $i$th is one. But this is exactly $\phi_\eta(z)$. And since $\phi_\eta(z) = \phi(z) + \Delta$, and the $i$th entry of $\phi(z)$ is also one, the $i$th entry of $\Delta$ must be zero. Similar reasoning applies to the cases when $z = \mathbf{c}_i, i \in \{2, ..., k\}$ or $z = \mathbf{c}_1$.

**Part 2.** Note that applying $I_{\eta,+}(\cdot)$ to the vector $\max(\mathbf{c}-z, 0)+\max(z-\delta-\mathbf{c}, 0)$ keeps all elements no more than $\eta$ and making all other elements one. As a result, $1-I_{\eta,+}(\max(\mathbf{c}-z, 0)+\max(z-\delta-\mathbf{c}, 0))$ would give a vector zero everywhere except those entries in $\max(\mathbf{c} - z, 0) + \max(z - \delta - \mathbf{c}, 0)$ which are $\leq \eta$. The set of indices which are $\leq \eta$ in the vector $\max(\mathbf{c} - z, 0) + \max(z - \delta - \mathbf{c}, 0)$ can be written as $\{j | j \in [k], 0 < z - \delta - \mathbf{c}_j \leq \eta, 0 < \mathbf{c}_j - z \leq \eta\}$, which is also the set of indices where $\phi_\eta(z)$ is nonzero. Since $\phi_\eta(z) = \phi(z) + \Delta$ and $\phi(z)$ has nonzero entries in the indices $\mathcal{I}$ and $\Delta$ has zero values at those entries by Part 1, then $\Delta$ must have nonzero entries at $\{j | j \notin \mathcal{I}, j \in [k], 0 < z - \delta - \mathbf{c}_j \leq \eta, 0 < \mathbf{c}_j - z \leq \eta\}$. □

### A.2.3 PROOF FOR THEOREM 1

**Theorem 1. Sparsity guarantee for FTA.** For any $z \in [l, u], \eta > 0$, $\phi_\eta(z)$ outputs a vector whose number of nonzero entries $\|\phi_\eta(z)\|_0$ satisfies:

$$\|\phi_\eta(z)\|_0 \leq 2\left\lfloor \frac{\eta}{\delta} \right\rfloor + 3$$

*Proof.* Similar to the proof of Theorem 1, we divide to three cases to prove the result.

**Case 1.** Consider the case that $\mathbf{c}_i < z < \mathbf{c}_{i+1}, i \in [k-1]$. Note that the number of nonzero entries in $\phi_\eta$ is equal to the number of entries less than $\eta$ in the vector equation 6, hence we can count the number of entries less than $\eta$ in $(z - \delta - \mathbf{c}_1, ..., z - \delta - \mathbf{c}_{i-1}, 0, \mathbf{c}_{i+1} - z, \mathbf{c}_{i+2} - z, ..., \mathbf{c}_k - z)$.

First, we count the number of entries that are less than or equal to $\eta$ on the left side of the $i$th position. Since the $i$th position is zero, which indicates $z - \delta - \mathbf{c}_i < 0$, hence $z - \delta - \mathbf{c}_{i-1} - \delta < 0$ and it follows that $0 < z - \delta - \mathbf{c}_{i-1} < \delta$. Then $\delta < z - \delta - \mathbf{c}_{i-1} + \delta = z - \delta - \mathbf{c}_{i-2} < 2\delta$. Hence $(j-1)\delta < z - \delta - \mathbf{c}_{i-j} < j\delta, j \in \{1, ..., i-1\}$. Assume there are $m$ entries $\leq \eta$ on the the left side of the $i$th. Then $z - \delta - \mathbf{c}_{i-m} \leq \eta$. It follows $(m-1)\delta < \eta$ and hence $m \leq \lfloor \frac{\eta}{\delta} \rfloor + 1$. Hence the total number of elements $\leq \eta$ on the left side of the $i$th position is at most $\lfloor \frac{\eta}{\delta} \rfloor + 1$.

Second, count the number of entries $\leq \eta$ on the right side of the $i$th position. Since $\mathbf{c}_i - z < 0$, $0 < \mathbf{c}_i + \delta - z = \mathbf{c}_{i+1} - z < \delta, 0 < \mathbf{c}_{i+2} - z < 2\delta, ....$ Hence the possible number of entries $\leq \eta$ on the right side of $i$th position is at most $\lfloor \frac{\eta}{\delta} \rfloor + 1$.

As a result, together with the $i$th position which is $0 \leq \eta$, the possible number of nonzero entries in this case is at most $2\lfloor \frac{\eta}{\delta} \rfloor + 3$.

**Case 2.** When $z = \mathbf{c}_i, i \in \{2, ..., k\}$, we count the number of entries less than $\eta$ in the vector

$$\max(\mathbf{c} - z, 0) + \max(z - \delta - \mathbf{c}, 0)$$
$$= ((i-1)\delta, ..., 2\delta, \delta, 0, 0, \delta, 2\delta, ..., (k-i)\delta)$$

Again, we attempt to count the number of entries less than $\eta$ in this vector by considering $(i-1)\delta, ..., 2\delta, \delta$ and $\delta, 2\delta, ..., (k-i)\delta$ respectively.

We follow the exactly same argument as above and now we have two zero entries at $i-1, i$th positions. The difference is that, the number of entries less than $\eta$ in the vector $((i-1)\delta, ..., 2\delta, \delta$ can be at most $\lfloor \frac{\eta}{\delta} \rfloor$. As a result, the number of nonzero entries in this case is still at most $2\lfloor \frac{\eta}{\delta} \rfloor + 2$.

**Case 3.** When $z = \mathbf{c}_1$, $\max(z - \delta - \mathbf{c}, 0)$ is a zero vector and $\max(\mathbf{c} - z, 0) + \max(z - \delta - \mathbf{c}, 0)$ is positive everywhere except the first entry, which is zero. Then we simply count the number of entries $\leq \eta$ on the right side of the 0th position, i.e. $j \in \{1, ..., k\}$. Similar to the analysis in the above Case 1, the possible number of entries $\leq \eta$ on the right side of 0th position is at most $\lfloor \frac{\eta}{\delta} \rfloor$. Hence in this case, the number of nonzero entries is at most $\lfloor \frac{\eta}{\delta} \rfloor + 1$.

In summary, the number of nonzero entries does not exceed $2\lfloor \frac{\eta}{\delta} \rfloor + 3$. This completes the proof. □

**Remark.** As we empirically demonstrated in Table 1 and below Figure 10 and Figure 9, the actual sparsity achieved by FTA is lower than the upper bound. This is because our upper bound is for any possible input $z$. Consider that in Case 1 in the above proof, we count the number of entries that

are less than or equal to $\eta$ on the left side and right side of the $i$th position. There are $\lfloor \frac{\eta}{\delta} \rfloor + 1$ such entries on both sides only when $z$ is exactly equal to $\frac{\mathbf{c}_i + \mathbf{c}_{i-1}}{2}$, which is unlikely to happen in practice.

### A.2.4  PROOF FOR COROLLARY 1

**Corollary 1** *Let $\rho \in [0, 1)$ be the desired sparsity level: the maximum proportion of nonzero entries of $\phi_\eta(z), \forall z \in [l, u]$. Assume $\rho k \geq 3$, i.e., some inputs have three active indices or more (even with $\eta = 0$, this minimal active number is 2). Then $\eta$ should be chosen such that*

$$\left\lfloor \frac{\eta}{\delta} \right\rfloor \leq \frac{k\rho - 3}{2} \quad \text{or equivalently} \quad \eta \leq \frac{\delta}{2} (\lfloor k\rho \rfloor - 1) \tag{7}$$

*Proof.* Because $\|\phi_\eta(z)\|_0 \leq \lfloor k\rho \rfloor$, from Theorem 1 it is sufficient to pick $\eta$ such that $2\lfloor \frac{\eta}{\delta} \rfloor + 3 \leq \lfloor k\rho \rfloor \leq k\rho$. This gives $\lfloor \frac{\eta}{\delta} \rfloor \leq (\lfloor k\rho \rfloor - 3)/2 \leq (k\rho - 3)/2$. Additionally, we know $\frac{\eta}{\delta} - 1 \leq \lfloor \frac{\eta}{\delta} \rfloor \leq (\lfloor k\rho \rfloor - 3)/2$, giving the second inequality. $\square$

### A.3  MORE DISCUSSION ABOUT FTA

Our development of the FTA assumed that the inputs are bounded in the range $[l, u]$ (recall that $\mathbf{c} = [l, l + \delta, l + 2\delta, ..., u]$). This is not guaranteed for the standard inputs to activations, namely $z = \mathbf{x}^\top \mathbf{w}$ for some weight vector $\mathbf{w}$. The FTA can still be used if $z \notin [l, u]$, but then the gradient of the FTA will be zero. This means that the weights $\mathbf{w}$ cannot be adjusted for inputs where $z$ becomes too large or too small. This issue is usually called *gradient vanish*, which is common in many popular, existing activation functions such as ReLU, tanh, sigmoid, etc.

In our main paper, we proposed to use different tiling vectors (i.e., $\mathbf{c}$s) with a broad range of bounds to different components in the input vector of FTA to reduce the chance of vanishing gradients. Here we discuss two other possible ways to avoid this issue: 1) use a squashing activation before handing $z$ to the FTA and 2) regularize (penalize) $z$ that falls outside the range $[l, u]$. For example, tanh can be applied first to produce $z = \tanh(\mathbf{x}^\top \mathbf{w}) \in [-1, 1]$. Though the simplest strategy, using tanh function can be problematic in deep neural networks due to gradient vanishing problems. An alternative is to use a penalty, that pushes out-of-boundary $z$ back into the chosen range $[-u, u]$

$$r(z) \stackrel{\text{def}}{=} I(|z| > u) \circ |z| \tag{8}$$

This penalty is easily added to the loss, giving gradient $\frac{\partial r(z)}{\partial z} = I(z > u) - I(z < -u)$. For example, $z = \max(\mathbf{x}^\top \mathbf{w}, 0)$, which might produce a value greater than $u$. If $z > u$, then the gradient pushes the weights $\mathbf{w}$ to decrease $z$. It should be noted that the number of nonzero entries can only decrease when $z$ goes out of boundary. However, in our experiments in our main paper, we found such out of boundary loss is unnecessary on all of our tested domains. Furthermore, we further verify that our FTA performs stably across different weights for the above regularization in Section A.5.

### A.4  REPRODUCING EXPERIMENTS FROM SECTION 5

**Common settings.** All discrete action domains are from OpenAI Gym (Brockman et al., 2016) with version 0.14.0. Deep learning implementation is based on tensorflow with version 1.13.0 (Abadi & et. al, 2015). We use Adam optimizer (Kingma & Ba, 2015), Xavier initializer (Glorot & Bengio, 2010), mini-batch size $b = 64$, buffer size 100k, and discount rate $\gamma = 0.99$ across all experiments. We evaluate each algorithm every 1k training/environment time steps.

**Algorithmic details.** We use $64 \times 64$ ReLU units neural network for DQN and $200 \times 100$ ReLU units neural network for DDPG. All activation functions are ReLU except: the output layer of the $Q$ value is linear. The weights in output layers were initialized from a uniform distribution $[-0.003, 0.003]$. Note that we keep the same FTA setting across *all* experiments: we set $[l, u] = [-20, 20]$; we set $\delta = \eta = 2.0$, $\mathbf{c} = \{-20, -18, -16, ..., 18\}$, and hence $k = 40/2 = 20$. This indicates that the DQN-Large and DDPG-Large versions have $64 \times 20 = 1280$ and $100 \times 20 = 2000$ ReLU units in the second hidden layer. For RBF coding, we set the bandwidth as $\sigma = 2.0$, and uses the same tiling (i.e. $\mathbf{c}$ vector) as our FTA.

**Meta-parameter details.** For DQN, the learning rate is 0.0001 and the target network is updated every 1k steps. For DDPG, the target network moving rate is 0.001 and the actor network learning rate is 0.0001, critic network learning rate is 0.001. For $l_1, l_2$ regularization variants, we optimize its regularization weight from $\{0.1, 0.01, 0.001, 0.0001\}$ on MountainCar, then we fix the chosen optimal weight 0.01 across all domains.

**Environmental details on discrete control domains.** We set the episode length limit as 2000 for MountainCar and keep all other episode limit as default settings. We use warm-up steps 5000 for populating the experience replay buffer before training. Exploration noise is 0.1 without decaying. During policy evaluation, we keep a small noise $\epsilon = 0.05$ when taking action.

**Environmental details on continuous control domains.** On Mujoco domains, we use default settings for maximum episodic length. We use $5,000$ warm-up time steps to populate the experience replay buffer. The exploration noise is as suggested in the original paper by Lillicrap et al. (2016).

## A.5 Additional Experiments on Reinforcement Learning Problems

This section shows the following additional empirical results which cannot be put in the main body due to space limitations.

- A.5.1 empirically investigates the representation sparsity generated by FTA.
- A.5.2 shows the comparison with Tile Coding NNs (Ghiassian et al., 2020), which first tile code inputs before feeding them into a neural network.
- A.5.3 shows the empirical results of DQN-FTA using linear activation function with regularization weights $\in \{0.0, 0.01, 1.0\}$ for the out of bound loss in Eq 8.
- A.5.4 shows the sensitivity of FTA to the sparsity control parameter $\eta$ and tile width parameter $\delta$.
- A.5.5 shows the results when applying FTA to both hidden layers rather than just the second hidden layer in RL problems.
- A.5.6 shows gradient interference analysis result.
- A.5.7 shows the result of using FTA on an autonomous driving application.

### A.5.1 Empirically Measuring Sparsity

We also report two sparsity measures of the learned representations averaged across all steps in Table 1. We compute an estimate of sparsity by sampling a mini-batch of samples from experience replay buffer and taking the average of them. Instance sparsity corresponds to proportion of nonzero entries in the feature vector for each instance. Overlap sparsity French (1991) is defined as $overlap(\phi, \phi') = \sum_i I(\phi_i \neq 0) I(\phi'_i \neq 0)/(kd)$, given two $kd$-dimensional sparse vectors $\phi, \phi'$. Low overlap sparsity potentially indicates less feature interference between different input samples. Theorem 1 guarantees that the instance sparsity, with FTA, should be no more than 12.5% for our setting of $k = 40, \eta = \delta$; and should be no more than 25% when $k = 20, \eta = \delta$. In the table, we can see that FTA has lower (better) instance sparsity than the upper bound provided by the theorem. Further, FTA achieves the lowest instance and overlap sparsity among all baselines in an average sense.

Figure 9 and Figure 10 are corresponding to the learning curves as shown in Figure 6 in Section 5.3. We show learning curves of instance/overlap sparsity as a function of training steps for FTA, $l_1, l_2$ regularization and RBF. The instance sparsity is computed by randomly sampling a mini-batch of states from replay buffer and count the average number of nonzero entries in that mini-batch divided by feature dimension $kd$. The overlap sparsity is computed by randomly sampling two mini-batches of states from the replay buffer and count the average number of simultaneously activated (i.e. nonzero) entries in both mini-batches. From the sparsity learning curves, one can see that our FTA has very stable sparsity since the beginning of the learning, and the sparsity is almost constant cross domains. Particularly, the overlap sparsity is very low, which possibly indicates low representation interference.

### A.5.2 Comparison with TC-NN

We include the figure of comparing with TC-NN from the work by Ghiassian et al. (2020), which use a regular tile coding to process the input before feeding into the neural network. It should be noted

Table 1: Sparsity on LunarLander (average across time steps)

| Sparsity | FTA(k=40) | FTA(k=20) | L1 | L2 | RBF |
|---|---|---|---|---|---|
| Instance | 7% | 14% | 16% | 44% | 99% |
| Overlap | 4% | 8% | 10% | 34% | 99% |

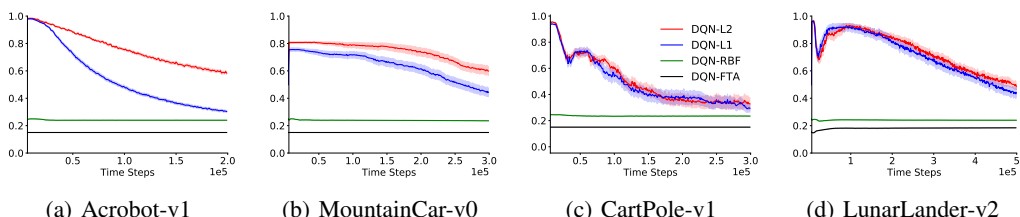

(a) Acrobot-v1     (b) MountainCar-v0     (c) CartPole-v1     (d) LunarLander-v2

Figure 9: Instance sparsity v.s. number of time steps on MountainCar, CartPole, Acrobot, LunarLander. The results are averaged over 20 random seeds and the shade indicates standard error.

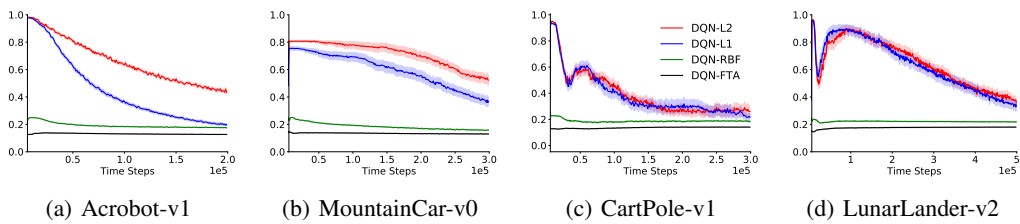

(a) Acrobot-v1     (b) MountainCar-v0     (c) CartPole-v1     (d) LunarLander-v2

Figure 10: Overlap sparsity (number of simultaneously activated entries in the sparse feature vectors) v.s. number of time steps by averaging over 20 random seeds and the shade indicates standard error.

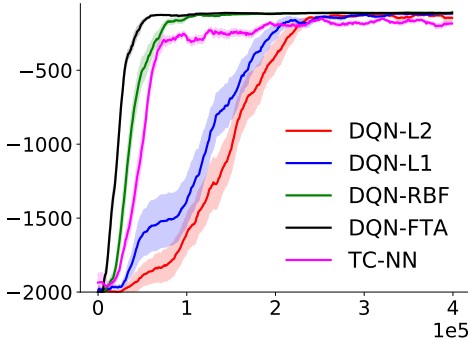

Figure 11: DQN-FTA compares with TCNN. Evaluation learning curves are averaged over 20 random seeds and the shade indicates standard error. All variants are trained without target networks.

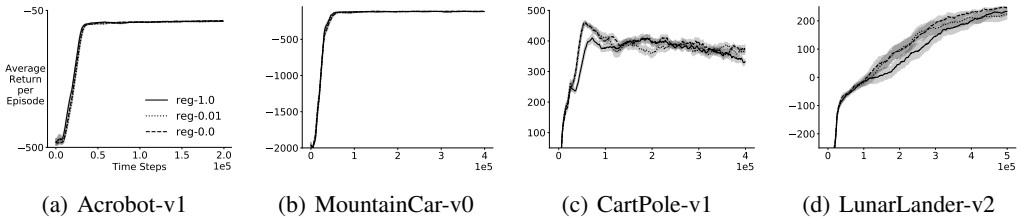

(a) Acrobot-v1  (b) MountainCar-v0  (c) CartPole-v1  (d) LunarLander-v2

Figure 12: DQN-FTA trained with linear activation function with different regularization weight (-reg-1.0 means regularization is set as 1.0). One can see that our FTA is *not sensitive to this choice* as those learning curves are almost overlapping with each other. Evaluation learning curves are averaged over 10 random seeds and the shade indicates standard error. All variants are trained without target networks.

that this method itself is not a sparse representation learning technique, and requires the original observation space to be scaled within $[0, 1]$. Figure 11 shows that TC-NN does indeed improve performance for DQN, but not compared to FTA. DQN-FTA both learns faster and reaches a better, more stable solution. In our TC-NN implementation, as a correspondent to FTA, we use binning operation to turn each raw input variable to 20 binary variables through binning operation. We use the same learning rate 0.0001 as other baselines. The result is generated by using FTA with a single tiling: $[l, u] = [-20, 20]$, $\delta = \eta = 2.0$, and hence $\mathbf{c} = \{-20, -18, -16, ..., 18\}$, $k = 40/2 = 20$.

### A.5.3 INSENSITIVITY TO OUT OF BOUNDARY LOSS

We demonstrate the effect of using an out of boundary loss as discussed in Section A.3. This is supplementary to our experiments in the main body, where we did not use an out of boundary loss, i.e. the regularization weight is 0. To highlight the issue of out of boundary, we intentionally use a tiling vector $\mathbf{c}$ with a small range $[-1, 1]$ with 20 tiles, i.e. $\delta = 2/20 = 0.1$ and set the sparsity control parameter $\eta = \delta = 0.1$. In Figure 12, one can see that our algorithm typically does not need such a regularization even if we use a small tiling.

### A.5.4 FTA'S STRONG PERFORMANCE WITH DIFFERENT TILE WIDTHS AND SPARSITY CONTROL PARAMETERS

The purpose of this section is to show that given an appropriate bound of the tiling vector $\mathbf{c}$, our approach is insensitive to tile width $\delta$ and sparsity control parameter $\eta$. Since we already showed in Section 5 that DQN-FTA works well on LunarLander with tiling vector bound $[-1, 1]$, we fix using this setting in this section. For both $\eta$ and $\delta$, we sweep over $0.8/2^i$, $i \in \{0, 1, 2, ..., 8\}$. Note that this range is extreme ($\delta = 0.8$ gives two bins, and $\eta = 0.8$ significantly increases overlap); we do so to see a clear pattern.

Figure 13 shows the early learning performance of all possible $9 \times 9 = 81$ combinations of $\eta, \delta$. We report the average episodic return (rounded to integers) within the first 300k time steps, for each combination. The results are averaged over 5 runs. The pattern is as expected. 1) The algorithms performs best with a reasonably small $\delta$ and $\eta$. 2) For extremely small $\eta$ and $\delta$, performance is poor, as expected because the gradient is very small and so learning is slow. 3) Given a fixed tile width $\delta$, the performance degrades as $\eta$ becomes smaller than $\delta$, since again the activation has many regions with zero derivative. 4) If $\eta$ gets too large, the sparsity level increases and again performance degrades, though performance remained quite good for a broad range between 0.025 and 0.2. 5) If $\delta$ gets large (big bins), performance degrades more so that with larger $\eta$, which matches the intuition that $\eta$ provides overlap rather than just increasing bin size and so losing precision.

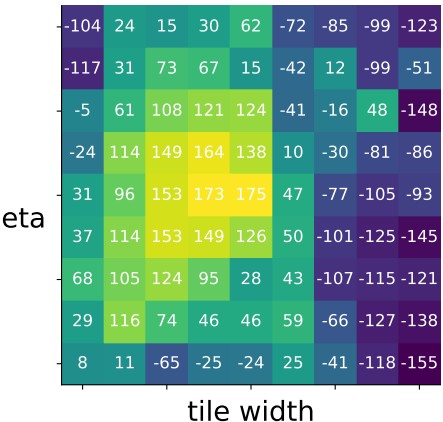

Figure 13: Sensitivity of $\eta, \delta$ on LunarLander-v2.

### A.5.5 EMPIRICAL RESULTS WHEN USING FTA TO BOTH HIDDEN LAYERS

As an activation function, FTA can be naturally applied in any hidden layer in a NN. As a convention, we typically use the same activation functions in a fully connected NN. In this section, we present such results on the reinforcement learning domains, as a supplement to the results in Section 5. All setting is the same as those in Section 5, except that we apply FTA to both hidden layers in DQN-FTA and DDPG-FTA. It should be noted that, in this case, DQN-FTA and DDPG-FTA have the same number of training parameters as their -Large correspondents respectively. Figure 14 shows the results on the discrete domains, and Figure 15 shows the results on the continuous control domains. It can be seen that, the FTA versions are better than ReLu versions in the case of not using a target network. This further validates the utility of our FTA in dealing with nonstationary problems.

### A.5.6 GRADIENT INTERFERENCE ANALYSIS

We provide gradient interference analysis in this section. The results in this section are produced by using FTA with a single tiling: $[l, u] = [-20, 20]$, $\delta = \eta = 2.0$, and hence $\mathbf{c} = \{-20, -18, -16, ..., 18\}$, $k = 40/2 = 20$.

We consider three measures for gradient interference. Let $l_\theta$ be the DQN loss parameterized by $\theta$. Given two random samples (i.e. experiences) $X, X'$ from ER buffer, we estimate

1. m1: $\mathbb{E}[\nabla_\theta l_\theta(X)^\top \nabla_\theta l_\theta(X')]$, this is to measure on average, whether the algorithm generalizes positively or interfere.

2. m2: $\mathbb{E}[\nabla_\theta l_\theta(X)^\top \nabla_\theta l_\theta(X')]$ ONLY for those pairs of gradient vectors who have negative inner product. This is to check for those likely interfered gradient directions, how much they interfere with each other.

3. m3: Within a minibatch of pairs of $(X, X')$, the proportion of negative gradient inner products (i.e. if 32 out of 64 pairs has negative inner products, then the proportion is 50%).

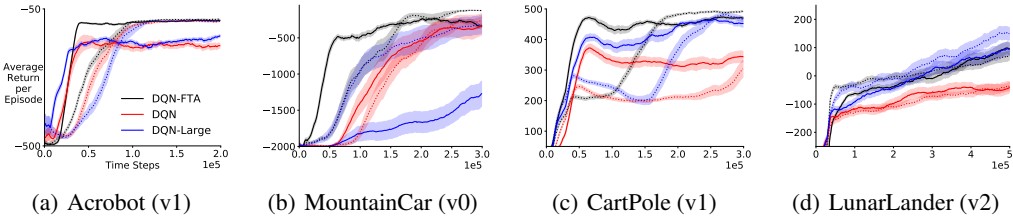

(a) Acrobot (v1)    (b) MountainCar (v0)    (c) CartPole (v1)    (d) LunarLander (v2)

Figure 14: Evaluation learning curves of of **DQN-FTA(black)**, **DQN(red)**, and **DQN-Large(blue)**, showing episodic return versus environment time steps. The **dotted** line indicates algorithms trained *with* target networks. The results are averaged over 20 runs and the shading indicates standard error. The learning curve is smoothed over a window of size 10 before averaging across runs.

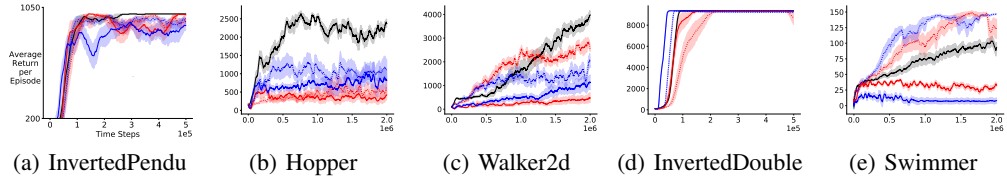

(a) InvertedPendu    (b) Hopper    (c) Walker2d    (d) InvertedDouble    (e) Swimmer

Figure 15: Evaluation learning curves of **DDPG-FTA(black)**, **DDPG(red)**, and **DDPG-Large(blue)** on Mujoco environments, averaged over 20 runs with shading indicating standard error. All algorithms are trained without target networks. The learning curve is smoothed over a window of size 30 before averaging across runs.

> This is to roughly check how likely it is to get conflict gradient directions by randomly drawing two experiences from ER buffer.

It should be noted that we normalize the gradient vectors to unit length before computing inner product. For each point in the learning curve, we randomly draw 64 samples from the ER buffer to estimate the interference.

Figure 17 shows the algorithms' performances in terms of number of time steps taken to reach a near-optimal policy: DQN-FTA > DQN-RBF **>** DQN-L1 > DQN-L2 $\geq$ DQN (i.e. > means better (use fewer time steps) and $\geq$ means slightly better).

There are several interesting observations. First, in Figure 16(b)(e), for the weights in the second hidden layer (m2 measurement), DQN-RBF has lower interference strength than DQN-FTA, but DQN-FTA performs better. This discrepancy indicates that it is not necessarily true that the lower interference, the better. Second, DQN tends to over generalize during early learning, which may hurt and result in bad performance. Third, figure (e) shows that DQN-L1 is similar to DQN-FTA during late learning stage in terms of m2. But DQN-L1 seems to have extremely interfered gradient directions during early learning, which may explain why it learns slower than DQN-FTA.

These observations may indicate that DQN-FTA helps generalize *appropriately*, but not overly generalize or highly interfered. We believe it is worth a separate work to thoroughly study the gradient interference issue to draw some interesting conclusions.

### A.5.7 TESTING STABILITY IN A SIMULATED AUTONOMOUS DRIVING DOMAIN

Our results have shown improved stability with FTA. In this section, we test FTA in an environment focused on stability, namely an autonomous driving task (Leurent, 2018). In the real world, a stable policy is of vital importance to ensure safety in autonomous driving. In this simulated task, the goal is not only to obtain high return, but also keep the number of car crashes as low as possible. FTA setting is the same as those in Section 5 on discrete control domains. Figure 18(a) shows the domain, where the agent—the green car—has to learn to switch lanes, avoid car crashes, and go as fast as possible. The observations are 25-dimensional, with vehicle dynamics that follow the Kinematic Bicycle Model. The action space is discrete.

FTA learns faster, with significantly fewer car crashes incurred during the evaluation time steps as shown in Figure 18(c). Target networks are harmful in this environment, potentially because they slow early learning; the agent to accumulate a significant number of crashes before improving.

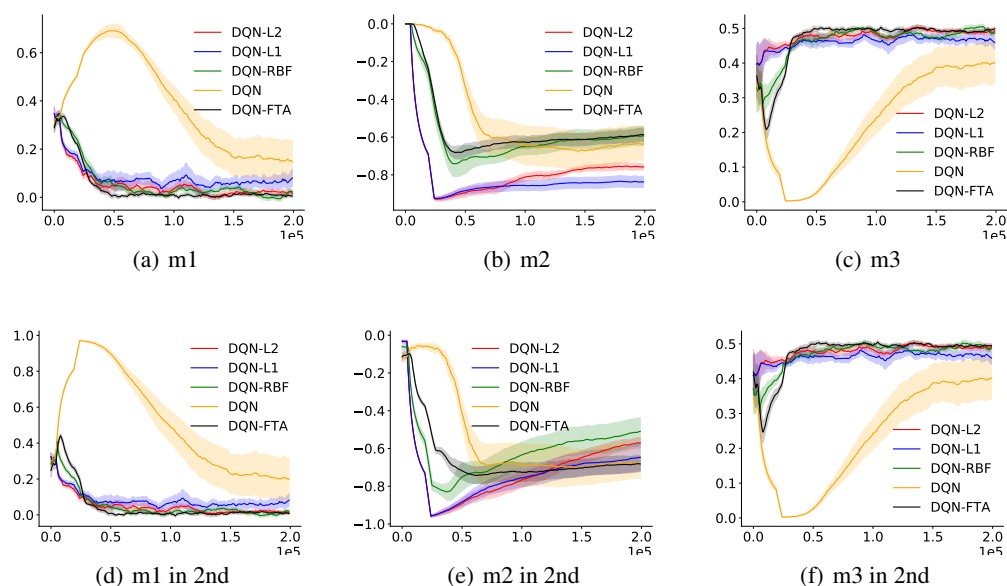

Figure 16: Gradient inference measurements on mountain car domains, averaging over 10 runs with the shade indicating standard error. (d)(e)(f) are measured for those parameters in the second hidden layer only (which are supposed to directly affect representation). All algorithms are trained without using target networks except DQN. The curve is smoothed over a window of size 20 before averaging across runs.

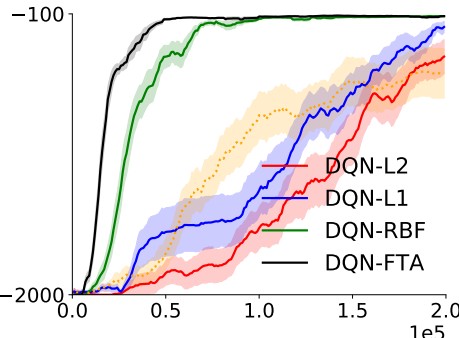

Figure 17: Evaluation learning curves on mountain car. The results are averaged over 20 runs with the shade indicating standard error. All algorithms are trained without using target networks.

## A.6 RESULTS ON IMAGE CLASSIFICATION TASKS

We now report the empirical results on two popular image classification tasks: MNIST (LeCun & Cortes, 2010) and Mnistfashion (Xiao et al., 2017). We found that FTA does not present any clear advantage or disadvantage in such a conventional supervised learning setting.

On MNIST, FTA achieves testing error $1.22\%$, and ReLu achieves $1.38\%$. On Mnistfashion, FTA achieves testing error $10.67\%$, and ReLu achieves $10.87\%$.

**Details.** The NN architecture we use is two convolution layer followed by two fully connected $32 \times 32$ layers. The first convolutional layer has 6 filters with size $5 \times 5$ and is followed by a max pooling operation. The second convolutional layer has 16 filters with the same size followed by a max pooling. FTA is applied to the second hidden layer and the setting is exactly the same as we used in the RL experiment A.4. To optimize learning rate, we optimize over the range

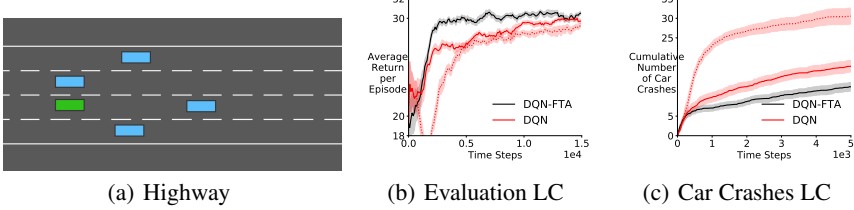

| (a) Highway | (b) Evaluation LC | (c) Car Crashes LC |

Figure 18: (a) The Highway environment. (b) The evaluation learning curve. (c) The cumulative number of car crashes as a function of driving time steps. Results are averaged over 30 runs with shading indicating standard error. The learning curve is smoothed over a window of size 10 before averaging across runs.

$\{0.00003, 0.00001, 0.0003, 0.0001, 0.003, 0.001\}$. We use 10-fold cross validation to choose the best learning rate and the above error rate is reported on testing set by using the optimal learning rate at the end of learning. The standard deviation of the testing error (randomness comes from NN initialization and random shuffling) is sufficiently small to get ignored.

### A.7 CONSTRUCTION OF PIECEWISE RANDOM WALK PROBLEM

In Section 4, recall that we train with data generating process $\{(X_t, Y_t)\}_{t \in \mathbb{N}}$ so that interference difficulty is controllable, while permitting fair comparison via a fixed equilibrium distribution.

The temporal correlation in high difficulty $\{X_t\}_{t \in \mathbb{N}}$ is designed to mimic state space trajectories through MDPs with a high degree of local state space connectedness. For example, an agent can only move to adjacent cells in GridWorld, the paddle and ball in Atari Breakout can only move so far in a single frame, and most successor states in Go only differ by a few pieces. Figure 19 depicts sample trajectories across a range of difficulties, alongside the fixed equilibrium distribution.

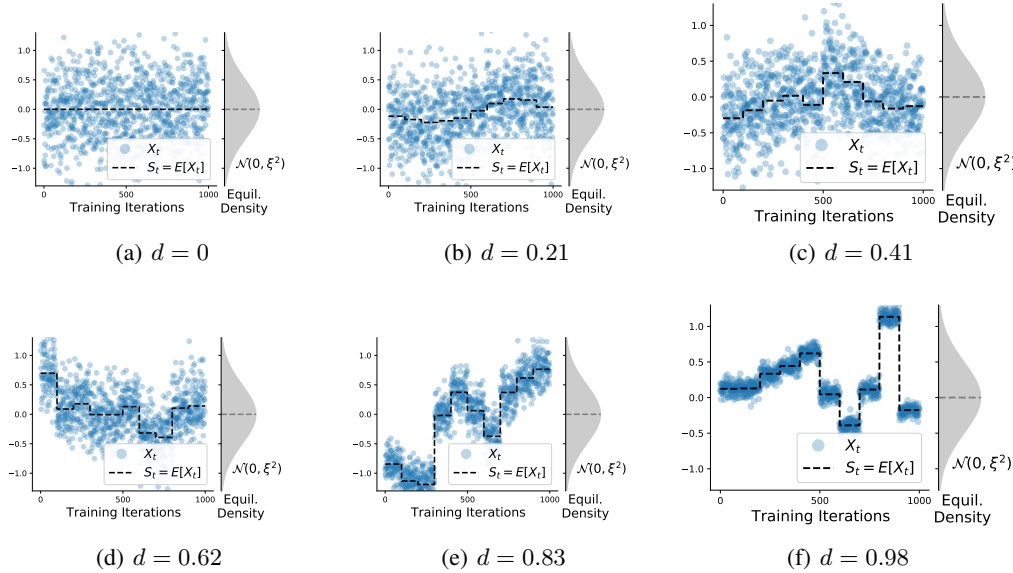

Figure 19: Sample trajectories of $\{X_t\}_{t \in \mathbb{N}}$ (blue) and $\{S_t\}_{t \in \mathbb{N}}$ (black) across a range of difficulties. Note in particular the fixed equilibrium distribution (gray) and i.i.d sampling for $d = 0$ (top left).

Here we rigorously construct $\{X_t\}_{t \in \mathbb{N}}$ and $\{S_t\}_{t \in \mathbb{N}}$, and show they have the claimed properties.

Let random variable $E_t \sim \mathcal{N}(\epsilon; 0, \sigma^2)$ be a source of Gaussian noise for the recursive random variable $S_t$

$$S_{t+1} = (1-c)S_t + E_t \qquad (9)$$

If $c \in (0,1]$, then $\{S_t\}_{t\in\mathbb{N}}$ is a linear Gaussian first order autoregressive process, denoted AR(1).[4] Then the *equilibrium distribution* of $\{S_t\}_{t\in\mathbb{N}}$ is also Gaussian. If $S_0$ is sampled from the equilibrium distribution, then the process $\{S_t\}_{t\in\mathbb{N}}$ is stationary with $E[S_t] = 0$ and variance

$$\nu^2 \doteq E[S_t^2] = \frac{\sigma^2}{2c - c^2} \qquad (10)$$

It also follows that $\{S_t\}_{t\in\mathbb{N}}$ is ergodic. See Grunwald et al. (1995) for reference on AR(1) processes, especially the discussion of equation (3.3) for linear Gaussian AR(1) processes, and section 5.3 for their stationarity and ergodicity.

Let $s_t$ be a realization of $S_t$. We define $X_t | S_t = s_t$ as a Gaussian r.v. with mean $s_t$ and variance $\beta^2$:

$$X_t | S_t = s_t \sim \mathcal{N}(x; s_t, \beta^2) \qquad (11)$$

The process $\{X_t\}_{t\in\mathbb{N}}$ is also a linear Gaussian AR(1) process, so we can again rely on established AR(1) process theory to conclude that the *equilibrium distribution* of $\{X_t\}_{t\in\mathbb{N}}$ is Gaussian. Moreover, if $X_0$ is sampled from the equilibrium distribution, then the process $\{X_t\}_{t\in\mathbb{N}}$ is stationary with $E[X_t] = 0$ and variance $\xi^2$

$$\xi^2 \doteq E[X_t^2] = \beta^2 + \nu^2 = \beta^2 + \frac{\sigma^2}{2c - c^2} \qquad (12)$$

Refer to Theorem 4 in Section A.7.3 for proof that $\{X_t\}_{t\in\mathbb{N}}$ is a linear Gaussian AR(1) process, and that the resulting equilibrium distribution has variance specified by equation 12. Note there are several variance terms in the above construction.

- $\sigma^2$ is the noise variance in the random walk, i.e. the source of jumps in $\{S_t\}_{t\in\mathbb{N}}$
- $\nu^2$ is the equilibrium distributon variance of $\{S_t\}_{t\in\mathbb{N}}$
- $\beta^2$ is the variance in any particular $X_t$ given $s_t$ at time step $t$
- $\xi^2$ is the equilibrium distribution variance of $\{X_t\}_{t\in\mathbb{N}}$

The processes $\{S_t\}_{t\in\mathbb{N}}$ and $\{X_t\}_{t\in\mathbb{N}}$ are fully determined by the quantities $c, \sigma^2, \beta^2$. For our experiments we wish to have a single parameter $d$ such that the equilibrium distribution of $\{X_t\}_{t\in\mathbb{N}}$ is fixed for all $d \in [0,1)$, but temporal correlation is variable. Also, $d = 0$ should correspond to zero temporal correlation. The following two theorems specify this behaviour more rigorously.

**Theorem 2.** *Let difficulty parameter $d \in [0,1)$ and $\xi^2 = (\frac{B}{2})^2$ for some $B \in \mathbb{R}^+$. Then $\{X_t\}_{t\in\mathbb{N}}$ will have fixed equilibrium distribution $\mathcal{N}(0, \xi^2)$, invariant to $d$, if parameters $c, \sigma^2, \beta^2$ are set as follows*

$$c = 1 - \sqrt{1-d}$$
$$\sigma^2 = d^2 (\frac{B}{2})^2$$
$$\beta^2 = (1-d)(\frac{B}{2})^2$$

$\xi^2$ is defined in terms of $B \in \mathbb{R}^+$ simply because $B$ is a more intuitive design parameter. In particular, $B$ is a high probability bound with $P(X_t \in [-B, B]) \leq 0.95$ for all $t$, since $\{X_t\}_{t\in\mathbb{N}}$ is ergodic.

**Theorem 3.** *$d = 0$ and $S_0 = 0$ induces i.i.d $X_t$ from $\mathcal{N}(x; 0, \xi^2)$, the equilibrium distribution of $\{X_t\}_{t\in\mathbb{N}}$.*

---

[4]If $c = 0$, then $S_t$ is a simple Gaussian random walk. We do not use such a process, because sample paths are unbounded, hitting $\pm\sqrt{t}$ infinitely often as $t \to \infty$

We defer proof of both theorems to Section A.7.3.

Correlation difficulty $d$ is intuitively characterized as follows.

- As $d$ increases, $\beta^2$ decreases, so a smaller portion of the overall state space is supported at any given time. i.e. $P(X_t = x | S_t = s_t)$ becomes increasingly narrow, inducing higher temporal correlation.

- As $d$ increases, the noise in random walk $\{S_t\}_{t\in\mathbb{N}}$ increases in amplitude, so that larger jumps in the mean of any particular $X_t$ are more likely.

- At $d = 0$, all correlation difficulty from $\{X_t\}_{t\in\mathbb{N}}$ has been removed, in that we recover iid sampling.

- As $d \to 1$, $\{X_t\}_{t\in\mathbb{N}}$ converges towards pathological correlation, with $X_t \xrightarrow{d} \delta(s_t)$, where $\delta(\cdot)$ is the Dirac delta distribution. i.e. $X_t$ becomes constant everywhere except the jumps in realization $s_t$ of $S_t$.

Despite the above relationships between $d$ and the other process parameters, the equilibrium distribution $\mathcal{N}(x; 0, \xi^2)$ is identical for all difficulty settings $d \in [0, 1)$, because the increase (or decrease) in parameters $\sigma^2, c$ is tuned specifically to counteract the decrease (or increase) in $\beta^2$. Having identical equilibrium distributions means that a hypothetical ideal algorithm, capable of perfectly mitigating interference, would train the identical approximator $f_\theta$ for any correlation difficulty $d \in [0, 1)$. Hence, we use approximation loss on the equilibrium distribution as a measure of robustness to interference.

To summarize, correlation difficulty $d \in [0, 1)$ controls the likelihood of interference-inducing samples throughout training, but also permits fair comparison between values of $d$ by the (fixed) equilibrium distribution of $\{X_t\}_{t\in\mathbb{N}}$.

### A.7.1    PIECEWISE RANDOM WALK ADDITIONAL PLOTS

Figure 20 depicts learning curves of loss over training time at different levels of correlation difficulty. Figure 21 shows learning rate sensitivity curves for the FTA and ReLU networks for three different difficulty settings. The ADAM optimizer Kingma & Ba (2015) was used in all experiments.

For the experiment in Figure 3, where one sample from each $X_t$ is used for each weight update, FTA outperforms ReLU even on i.i.d data ($d = 0$). In order to find the conditions under which ReLU and FTA both perform equally well, we repeat the same experiment, but with 50 samples drawn from each $X_t$. Figure 23 depicts the results, where the performance is indeed equalized for iid sampling ($d = 0$), but ReLU's performance still diverges similarly to other experiments as correlation difficulty is increased.

The size of the FTA and ReLU neural networks used on the Piecewise Random Walk Problem were chosen based on best performance on iid data, and the best performers do not have the same number of parameters. (67K vs 5.2K learnable parameters.) In order to verify that the difference in parameter count was not a significant factor in handling high correlation difficulty, we additionally include results for a ReLU network with wider hidden layers totalling 81K learnable parameters. See Figure 22. Both ReLU networks perform very similarly—in most cases agreeing within $p = 0.05$—with loss steeply running away as correlation difficulty increases. So we conclude FTA's robustness to high correlation difficulties is not explained simply by parameter count.

### A.7.2    PIECEWISE RANDOM WALK PROBLEM HYPERPARAMETER SELECTION

**Experimental Parameters**

Configuration for Piecewise Random Walk problem across all experiments:

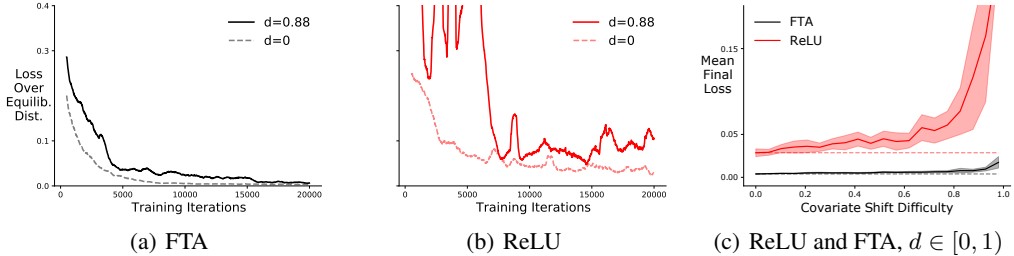

(a) FTA  (b) ReLU  (c) ReLU and FTA, $d \in [0, 1)$

Figure 20: Left, Middle: Learning curve of loss over stationary distribution during training on low difficulty (dotted) and high difficulty (solid) settings for two layer neural nets. The curves are smoothed over a window of 50. Right: The final loss over stationary distribution after 20K training iterations across a range of difficulty settings, shown as the mean of 30 runs with the shaded region corresponding to $p = 0.001$. The final loss per run is computed as the mean over the final 2.5K iterations, with the iid setting $d = 0$ (dotted) shown for baseline comparison.

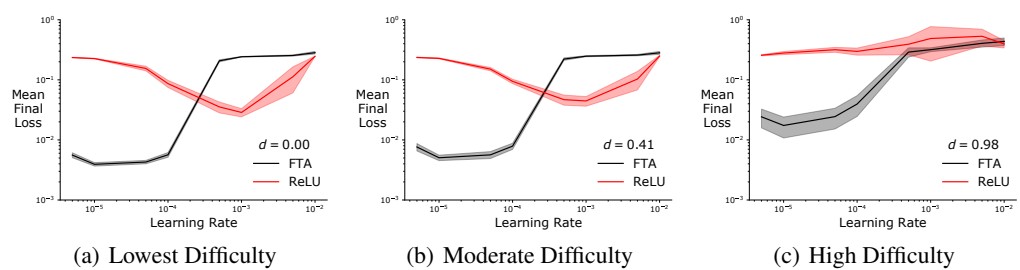

(a) Lowest Difficulty  (b) Moderate Difficulty  (c) High Difficulty

Figure 21: Learning rate sensitivity of FTA and ReLU for iid, mildly correlated, and severely correlated $X_t$ (left, middle, right, respectively.) Final loss performance is shown as the mean of 30 runs with the shaded region corresponding $p = 0.001$. These curves corroborate our findings that, in general, FTA prefers lower learning rates (but converges more quickly nonetheless.)

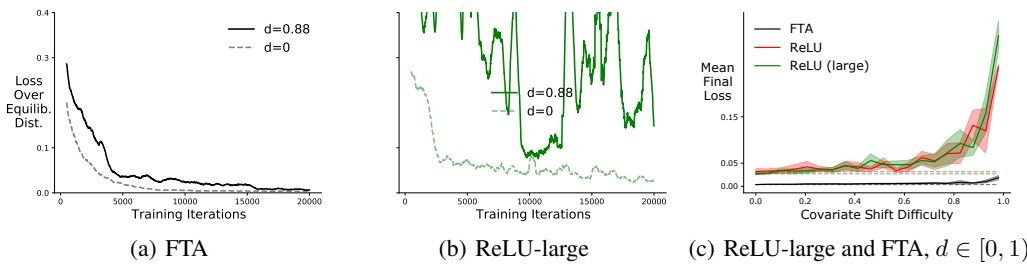

(a) FTA  (b) ReLU-large  (c) ReLU-large and FTA, $d \in [0, 1)$

Figure 22: Left, Middle: Learning curve for single run using loss over stationary distribution during training on low difficulty (dotted) and high difficulty (solid) settings for two layer neural nets, both having similar numbers of learnable parameters. The curves are smoothed over a window of 50. Right: The final loss over stationary distribution after 20K training iterations across a range of difficulty settings, shown as the mean of 10 runs with the shaded region corresponding $p = 0.05$. The final loss per run is computed as the mean over the final 2.5K iterations, with the iid setting $d = 0$ (dotted) shown for baseline comparison.

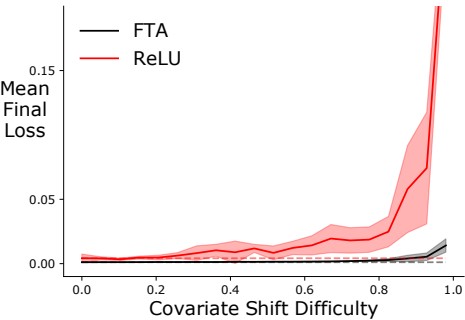

Figure 23: The final loss over stationary distribution after 20K training iterations across a range of difficulty settings, shown as the mean of 10 runs with the shaded region corresponding $p = 0.001$. For each weight update, a batch of 50 samples are drawn from each $X_t$, rather than a single sample as was done in other similar figures.

| | |
|---|---|
| Iterations per training run | 20K |
| Test batch size per iteration | 100 |
| Train batch size per iteration | 1 |
| Min, max difficulty settings $d$ | (0.0, 0.98) |
| Number of difficulty settings swept | 20 (linear range over (min, max), inclusive) |
| Training runs per difficulty setting | 10 for ReLU Large (Figure 22) |
| | 10 for batched run (Figure 23) |
| | 30 otherwise |
| Number of piecewise stationary segments | 50 |
| Target function | $Y_t = \sin(2\pi X_t^2)$ |
| High probability bound $B$ | $(-1, 1)$ |

Configuration for the neural networks used on the Piecewise Random Walk problem:

**FTA**

| | |
|---|---|
| Hidden layers | 2 |
| Layer width $w$ | 40 |
| Layer binning interval bounds $(l, u)$ | $(-1, 1)$ |
| Layer bins $k$ | 40 |
| Layer sparsity parameter $\eta$ | $\frac{1}{40}$ |

**ReLU**

| | |
|---|---|
| Hidden layers | 2 |
| Layer width $w$ | 50 |

**ReLU Large**

| | |
|---|---|
| Hidden layers | 2 |
| Layer width $w$ | 200 |

Layer widths $w$, bin count $k$, and sparsity parameter $\eta$ were chosen by sweeping a range, and choosing the best performing configuration on the iid setting (i.e. difficulty $d = 0$). In the sweep, the same $w, k, \eta$ are used for each hidden layer of FTA, and the same $w$ is used for each hidden layer of ReLU. See the Parameter Selection Details below. For all three networks, the ADAM optimizer Kingma & Ba (2015) was used with parameters $(\beta_1, \beta_2) = (0.9, 0.999)$, and the following learning rates were swept for each difficulty setting.

Learning rates $\lambda$    0.01, 0.005, 0.001, 0.0005, 0.0001, 0.00005, 0.00001, 0.000005

The results from the best performing learning rate are reported in Figures 3(c), 20(c), 22(c), and 23. That is, the performing learning rate is determined separately for each difficulty setting, using the mean final loss across 30 runs for the FTA and ReLU networks. 10 runs were used for ReLU Large and the batching run. Results from all learning rates are reported in Figure 21 for three different difficulty settings.

**Parameter Selection Details**

Before sweeping any difficulty settings, first the network architectures were established for the FTA and ReLU networks so that loss was minimized on the iid setting ($d = 0$). We avoid the overparametrized regime, so that standard bias-variance tradeoff applies here Belkin et al. (2019). (However, the ReLU Large network is overparametrized.) Learning rate was also swept during the architecture sweep, to ensure that the chosen architecture performed best within the learning rate range chosen for the main experiment depicted in Figure 3(c).

For the 2 layer FTA network, each hidden layer uses the same width $w$, and sparsity parameter $\eta$. Those parameters, along with learning rate $\lambda$ were optimized in the following ranges with grid search:

- Layer width $w \in \{10, 15, 20, 30, 40\}$
- Sparsity parameter $\eta = \frac{1}{w}$
- Learning rate $\lambda \in \{0.0005, 0.0001, 0.00005, 0.00001, 0.000005\}$

For the 2 layer ReLU network, fully connected hidden layers were used. Width $w$ and learning rate $\lambda$ were found from the following ranges, also with grid search:

- Layer width $w \in \{5, 10, 20, 30, 40, 50, 60, 70, 80, 100, 120\}$
- Learning rate $\lambda \in \{0.005, 0.001, 0.0005, 0.0001, 0.00005\}$

The learning rates are of different magnitude between FTA and ReLU because they perform best in different ranges, as explained in A.7.1. For the main experiments, where a range of difficulties are swept, the range of tested learning rates is shared between FTA and ReLU, and is broad enough to cover the optimal range for both. Also note that we set $\eta = \frac{1}{k}$, rather than separately sweeping it. This is because FTA performs consistently well with $\eta$ in this order of magnitude.

### A.7.3 PROOFS FOR COVARIATE SHIFT PROPERTIES

**Theorem 4.** *Let* $\{E_t\}_{t\in\mathbb{N}}, \{S_t\}_{t\in\mathbb{N}}, \{X_t\}_{t\in\mathbb{N}}$ *be stochastic processes such that*

$$E_t \sim \mathcal{N}(\epsilon; 0, \sigma^2)$$
$$S_{t+1} = (1-c)S_t + E_t \tag{13}$$
$$X_t | S_t = s_t \sim \mathcal{N}(x; s_t, \beta^2) \tag{14}$$

*where* $c \in (0, 1]$ *and* $s_t$ *is a realization of* $S_t$. *Then* $\{X_t\}_{t\in\mathbb{N}}$ *is a linear Gaussian AR(1) process with equilibrium distribution* $\mathcal{N}(x; 0, \beta^2 + \frac{\sigma^2}{2c-c^2})$.

*Proof.* Begin by rewriting equation 14 to give $X_t$ as a sum of $S_t$ and a mean zero Gaussian r.v. $B_t$

$$X_t = S_t + B_t \qquad\qquad B_t \sim \mathcal{N}(0, \beta^2), \text{ iid} \tag{15}$$

Now substitute according to equation 13

$$X_t = (1-c)S_{t-1} + E_{t-1} + B_t \tag{16}$$

Let Gaussian r.v. $\Theta \sim \mathcal{N}(0, \alpha^2)$ with some variance $\alpha^2$.

$E_{t-1}, B_t$ are independent Gaussian, so $E_{t-1} + B_t$ can be written as a linear combination of $B_{t-1}, \Theta_t$, since all are independent. Specifically, fix $\alpha^2$ so that the following holds

$$E_{t-1} + B_t = (1-c)B_{t-1} + \Theta_t \tag{17}$$

and substitute into equation 16

$$X_t = (1 - c)S_{t-1} + (1 - c)B_{t-1} + \Theta_t$$
$$= (1 - c)(S_{t-1} + B_{t-1}) + \Theta_t$$
$$= (1 - c)X_{t-1} + \Theta_t$$

By inspection, $\{X_t\}_{t \in \mathbb{N}}$ is a linear Gaussian AR(1) process with coefficient $(1-c)$ and noise variance $\alpha^2$. Elementary AR process theory gives the equilibrium distribution as

$$\mathcal{N}\left(0, \frac{\alpha^2}{1 - (1 - c)^2}\right) \tag{18}$$

For equation 17 to hold, we need the first and second moments of LHS and RHS to be equal. All terms are mean zero Gaussian, so it suffices to show when the LHS and RHS have equal variance:

$$\mathrm{Var}(E_{t-1} + B_t) = \mathrm{Var}((1 - c)B_{t-1} + \Theta_t)$$
$$\sigma^2 + \beta^2 = (1 - c)^2 \beta^2 + \alpha^2$$
$$\alpha^2 = \sigma^2 + \beta^2 - (1 - c)^2 \beta^2$$
$$\alpha^2 = \sigma^2 + (1 - (1 - c)^2)\beta^2$$

Substituting into equation 18

$$\mathcal{N}\left(0, \frac{\alpha^2}{1 - (1 - c)^2}\right)$$
$$= \mathcal{N}\left(0, \frac{\sigma^2 + (1 - (1 - c)^2)\beta^2}{1 - (1 - c)^2}\right)$$
$$= \mathcal{N}\left(0, \frac{\sigma^2}{1 - (1 - c)^2} + \beta^2\right)$$
$$= \mathcal{N}\left(0, \frac{\sigma^2}{2c - c^2} + \beta^2\right)$$

So $\{X_t\}_{t \in \mathbb{N}}$ is linear Gaussian AR(1) with the desired equilibrium distribution.

$\square$

**Theorem 2**

Let difficulty parameter $d \in [0, 1)$ and $\xi^2 = (\frac{B}{2})^2$ for some $B \in \mathbb{R}^+$. Then $\{X_t\}_{t \in \mathbb{N}}$ will have fixed equilibrium distribution $\mathcal{N}(0, \xi^2)$, invariant to $d$, if parameters $c, \sigma^2, \beta^2$ are set as follows

$$c = 1 - \sqrt{1 - d}$$
$$\sigma^2 = d^2 (\frac{B}{2})^2$$
$$\beta^2 = (1 - d)(\frac{B}{2})^2$$

*Proof.* By the above theorem 4, $\xi^2 = \beta^2 + \frac{\sigma^2}{2c - c^2} = (1 - d)(\frac{B}{2})^2 + d^2(\frac{B}{2})^2/d = (\frac{B}{2})^2$. This completes the proof. $\square$

**Theorem 3**

$d = 0$ and $S_0 = \delta(0)$ induces iid $X_t$ from $\mathcal{N}(x; 0, \xi^2)$, the equilibrium distribution of $\{X_t\}_{t \in \mathbb{N}}$.

*Proof.*

$$
\begin{aligned}
S_{t+1} &= (1 - c)S_t + E_t & \text{(by 9)} \\
&= S_t + E_t & (d = 0 \implies c = 0) \\
&= S_t & (d = 0 \implies \sigma^2 = 0 \implies E_t \sim \delta(0))
\end{aligned}
$$

Hence, $d = 0$ implies $S_t$ is constant over all time $t$, and $S_0 = \delta(0)$ gives $S_t = \delta(0)$ for all time $t$.

Let r.v. $B_t \sim \mathcal{N}(0, \beta^2)$ iid, then $X_t = S_t + B_t = B_t$ when $S_t$ is Dirac delta concentrating at zero. $\beta^2 = (1 - d)(\frac{B}{2})^2 = (\frac{B}{2})^2$ by the setting from Theorem 2 and setting $d = 0$. Hence, $d = 0$ induces iid sampling from the equilibrium distribution of $X_t$. This completes the proof. $\qquad\square$

