# OpenReview forum: "Fuzzy Tiling Activations: A Simple Approach to Learning Sparse Representations Online"
_ICLR.cc/2021/Conference — ICLR 2021 Poster_

### Official Review · AnonReviewer1 · 2020-10-27
**promising and simple idea, great paper**

**Rating:** 7
**Confidence:** 4

**Review:**

(I will keep my review short, as I don't have much constructive criticisms to share)

The authors propose a new activation function that guarantees sparsity, which can reduce interference/forgetting when learning on non-stationary data. The Leaky Tiling Activations (LTA) are relaxations of the binning operation to make it differentiable. The authors proved that LTA guarantees sparse features.

Next, LTA is experimentally evaluated in online supervised learning on non-stationary data, as well as in reinforcement learning (RL).  In the supervised learning experiment, LTA is shown to outperform ReLU on a toy task. In the RL experiments, the story is repeated in discrete action domains (with a DQN backbone) and in continuous action domains (with a DDPG backbone).

The paper is well written and an enjoyable read. It is well motivated given that sparsity is an important characteristic for learning on non-stationary data. The empirical section is extensive. I am not really knowledgeable of those experiments, however. This is why I chose a confidence score of 4.

My main criticism would be that no code was shared. I am willing to keep my score and fight for acceptance if the authors commit to releasing the code upon acceptance. I would have appreciated, however, that the codebase would be available during the review period. This is why I chose a score of 7 instead of 8.
Furthermore, I encourage the authors to build simple PyTorch and/or TF packages such that one can easily use the LTA without much overhead.

_________

**Post rebuttal**

I am happy with the response.

---

> ### Author Response · Authors · 2020-11-21
> **Response to R1**
>
> We thank you for reading our paper and providing positive and helpful feedback.
>
> Before the end of the discussion period, we will release the LTA code (in PyTorch), and this was used for the supervised learning experiment in Section 4. The RL code (in TensorFlow) is embedded in a larger repository and will take a bit more time to release, but we will release it before a camera-ready deadline if the paper can get accepted. As a note, the key component of the code is the LTA activation implementation, which can be easily used like other activation functions in TensorFlow. For example, in our code, this is written as hidden_layer = tf.layers.dense(previous_layer, 64, activation=LTA), which returns a tensor with shape (batch size, 64 * 20 = 1280).

---

### Official Review · AnonReviewer2 · 2020-10-27

**Rating:** 7
**Confidence:** 4

**Review:**

This paper proposes a novel activation function based on tiling, with a careful consideration of the tradeoff between differentiable regions of the tiling (which would otherwise be an undifferentiable one-hot) and sparsity. Using such activations in domains where we expect sparsity to help, online learning and deep RL, shows some encouraging improvements.

Strengths:
- The paper is fairly well written and easy to understand
- The method is simple and addresses important problems in deep RL
- Results are encouraging

Weaknesses [the two following weaknesses we addressed in the rebuttal]:
- The central motivation for this method is the reduction in gradient interference, yet, it is never measured, nor other similar values. Sparsity is measured, but it is expected to be lower, by _construction_. Proposing new methods without understanding why they work is detrimental to progress. "My number is bigger than your number" is not a great way to do science.
- The comparisons and baselines may not be entirely fair (although the methodology is otherwise good, with repeated runs and confidence intervals)

I gave the paper a score of 6 because while the proposed method is interesting and possibly useful, more needs to be done to understand it, and advance our understanding of deep RL.
[Rebuttal update: the extra results clear up some uncertainty about this method. I think the success of this method is interesting and teaches us a thing or two about deep RL.]


Comments:
- "past $X_{t−1}$, $X_{t−2}$", that would be a 2-Markov problem, and is needlessly specific? I'd suggest just writing "the past $X_{t-i}$" since general online problems can have arbitrary time-dependencies.
- "such encodings can enable faster learning and reduce interference", that may be true, but the authors need to test this.
- "LTA itself does not introduce any new training parameters", that's not true. If a layer has a 100 units and 4 LTA bins, then the output of LTA is 400 units, and the following layers has to be 4 times as big, and thus introduces 300 new training parameters.
  - "One potentially surprising point is that the LTA reaches a lower error, even on iid data", this should be a warning sign. When a new method outperforms an old method in settings where it shouldn't, it's often because the setting is unfair to the old method. I see in Figure 17 that the authors compared LTA to a wider ReLU network, that's good, although it's still not clear to me that the comparison is fair, as values of k and eta are not given, nor how many units and layers there are. It would be good to have all hyperparameters clearly laid out, especially in the appendix (which can be arbitrarily long).
  - (The RL section is much clearer in that regard!)
- Is the (presumed) reduction in gradient interference really due to the sparse encoding? or due to LTA still being mostly 0-gradient everywhere (e.g. in Figure 1c, with k=3 and eta=0.1, 60% of the x-axis has gradient 0)? This should be tested.
  - If it's not clear what I mean, let's say we have 100 LTA units with k=4 and so 400 outputs. By construction we're going to have at most a 25% sparsity, let's say for argument's sake that all LTA inputs are positive and so there exactly 25% or 100 units that are on. Some proportion of these 100 active units will be in the 0-gradient regime, even though they are active. If that proportion is 5%, then activation sparsity and gradient sparsity will be roughly the same, but if that proportion is 80%, then the gradients will be _even more_ sparse than the activations. That seems like something that's important to know to understand how/why the proposed method works.
- The RL experiments compare DQN-LTA with DQN-Large, this is good, but perhaps misleading
  - it's not clear how k/delta/eta were chosen (Figure 13 hints at good values, but the legend is missing and it's not clear if this was done post-hoc or if it informed the choice in the main experiments, I assume the latter)
  - it's not clear that the k that's optimal for DQN-LTA is the same that's optimal for DQN-Large. Considering Figure 13, it seems that the authors spent more time doing hyperparameter search for DQN-LTA than for DQN-Large, in that sense it's likely an unfair comparison. (in any case it would be interesting to see a plot similar to Figure 13 for DQN)

---

> ### Author Response · Authors · 2020-11-21
> **Response to R2**
>
> We thank you for your highly constructive and useful comments, they will make our paper stronger. We included missing details in our updated paper. Our response below includes references to the following PDF document (we could incorporate this into our appendix, but we upload here for your convenience):
> https://anonymousfiles.io/BYCDgCc5/
>
> “Central motivation is to reduce gradient interference”. Our central motivation is to design an effective online sparse representation learning method to reduce representation interference. We talk about obtaining sparse features to reduce interference when updating value estimates. For sparse representation acquired by LTA in the final hidden layer, measuring overlap sparsity should be the most intuitive/straightforward way to show the effect in reducing interference. Notice that when two inputs have no overlap (have different features active), then updates to weights on the output layer for one input have no impact on the values for the other input. When overlap sparsity is lower, therefore, we will have lower interference for value estimates.
>
> However, the implication of gradient interference is not quite clear. It is possible for the magnitude of the dot product of two gradient vectors to be large despite the overlap sparsity of the 2nd/final hidden layer is low. Then high interference measured in this way does not mean that the value estimates of some state-action pairs are more likely to get perturbed when updating other state-action values.
>
> Based on your request, we do provide gradient interference and include brief discussions.  Please look at Section 1 in the above PDF file. We believe it deserves a separate work to thoroughly study the gradient interference issues to draw some interesting conclusions, as this should at least involve addressing questions like 1. How the gradient interference should be measured; 2. what the appropriate level of gradient interference should be. Consider the extreme case where there is zero interference by any measure: the tabular setting. The RL agent will have to physically visit each state to get value estimates. This should have low sample efficiency on large domains. In general, it is not the case that the lower interference, the better.
>
> Related to the gradient interference is the point that the gradient could be zero in many places, resulting in interference reduction in an arguably undesirable way. But, this is not the case. Note that our LTA setting does not give a sparse gradient. We apologize that if Figure 1 in our paper misled you. Please see Section 2 in the above PDF file for detailed analysis.
>
> For more specific points:
>
> 1. “LTA itself does not introduce any new training parameters.” We meant that it does not introduce learning parameters to produce a larger number of features. You are absolutely right that the number of weights that use the LTA’s output has to be higher dimensional than a non-sparse representation. We considered this a separate point, since in general if you want to use a sparse representation, we should expect to have a higher dimensional feature. Consider a 64 * 64 two hidden layer network. Applying LTA on the second hidden layer leads to, say, 1280 weight units in the output layer. But the second hidden layer is still a 64 * 64 matrix and so LTA itself does not introduce any new training parameters. This is a nice property, as another approach to getting such a sparse layer might increase both the number of parameters to get the sparse layer and the output layer. For example, DQN-Large with L1/L2 needs a large weight matrix (64 * 1280) in the 2nd hidden layer to project to a high dimension to match the dimension of the output of LTA. Hence DQN-Large increases the number of training parameters in both the second hidden layer and the output layer. In any case, we will better qualify this statement.
>
> 2. “One potentially surprising point is that the LTA reaches a lower error, even on iid data”. Though it is surprising, it is not an indication of an error and we did try to determine why as explained in the paragraph with that sentence. We stated that the result is likely due to using individual samples, rather than mini-batches. LTA helps under this noisier setting. We have now tested the case where we use mini-batches, and included those (as well as missing details) in the appendix, and find that this is indeed the reason.
>
> 3. “The RL experiments compare DQN-LTA with DQN-Large, this is good, but perhaps misleading.”
>
> We did not actually use Figure 13 to select hyperparameters. We simply used what we thought to be reasonable LTA settings; we used the same LTA settings across all the experiments in Section 5. In Section 3 in the above PDF file, we have included all the learning curves of DQN-L1 with a different number of units in the second hidden layer.
>
> We would be very happy to discuss this further with you. Please let us know if you have further questions.

---

> > ### Comment · AnonReviewer2 · 2020-11-23
> > **Clarifications**
> >
> > Thank you for your clarifications, they make a lot of sense. I'm happy to raise my score to 7.
> >
> > On interference, it is true that we do not yet have a good notion of what is the right way to measure it. I still think it is relevant to report some version of it, and as you have found we do see differences from DQN and other regularized DQNs. Any version you choose will be informative of some aspect of interference. I do suggest adding these results to the appendix, perhaps you can present them as a sanity check or something of the sort, rather than a conclusive result.
> >
> > It would be interesting to see the robustness to hyperparameter choices of DQN-LTA, I suspect it is more so. It could be an interesting avenue also for future work.

---

> > > ### Author Response · Authors · 2020-11-24
> > > **Response to R2**
> > >
> > > We are glad to know that our clarification makes a lot of sense for you.
> > >
> > > Sure, we will add to the final version of the paper: 1) those gradient interference results, and 2) the results with different hyperparameter choices of DQN-LTA.

---

### Official Review · AnonReviewer3 · 2020-10-29
**Weak accept: interesting idea, could have benchmarked against stronger baselines, needs revision for better reproducibility**

**Rating:** 7
**Confidence:** 4

**Review:**

### Update during review period

- The reproducibility of the paper is now much better. It's great that the authors promised to release the LTA code. I hope that this includes the code for the experiments.
- Based on the above, I changed my review score to 7.

### Summary

The paper presents a novel activation function (Leaky Tiling Activation - LTA) to produce sparse activations, which have been found to stabilize learning in continual learning and RL settings. The new nonlinearity and its theoretical properties are described well, and the authors present convincing experiments demonstrating that the method yields practical benefits on synthetic datasets and RL games (e.g. Atari).

### Reasons for score:

The paper should certainly be published somewhere, but maybe in a workshop that focuses on continual learning or RL.

While the proposed new activation function may be useful in some settings, there is not enough evidence in the paper that it would become a go-to solution, or significantly change the way we think about interference in continual learning and RL. In particular, the authors compare LTA variants of DQN, but it might have been useful to compare with e.g. Rainbow too. While I agree that interference is still an issue in modern RL with function approximation, it would be useful for potential users of LTA to know whether LTA provides benefits when used in conjunction with an existing state-of-the-art RL algorithm. Adding an experiment to this effect to the appendix would make this paper stronger.

### Pros

1. The paper addresses a key problem in continuous learning and RL: interference and catastrophic forgetting. It presents a novel method to combat this problem, and demonstrates its usefulness in a number of experiments.
2. The paper is generally well-written.
3. The experiments on synthetic data were compelling, and made for a very nice controlled experiment.


### Cons

1. The authors could have benchmarked against stronger baselines. The authors might be able to do this during the review process.
2. The precise set-ups that the authors used for their experiments should be described more clearly. As it is, the paper's work is not reproducible. See my the section "Questions during rebuttal period" below for details.

### Questions during rebuttal period:

Have the authors considered to benchmark using stronger (state-of-the-art) baselines? Someone who considers using LTAs would likely use a state-of-the-art method already, not DQN, which the authors benchmarked against. The question is then whether LTA yields benefits when used in conjunction with state-of-the-art methods, not when used in conjunction with DQN.

I did not fully understand what architecture the authors used in their experiments. The architecture appears to be described mostly in this single sentence: “All the algorithms use a two-layer neural network, with the primary difference being the activation used on the last layer. ”
- Between what and what is this difference? Between baselines and the LTA experiments?
- Later on the page from which I quoted above, the authors mention that they experimented with DQN-like networks. The original DQN paper used 3 layers (https://www.cs.toronto.edu/~vmnih/docs/dqn.pdf). Did the authors mean to write ““All the algorithms use a two-layer neural network”?
- Do the authors insert the LTA before the third layer?

More generally, it would be helpful if the authors could describe the architectures used in more detail and maybe ask a colleague who is not yet familiar with the paper to review for clarity and reproducibility.

### Some suggestions

I would rephrase the first sentence of the abstract in order to introduce “interference” in a gentler way. While interference is an active research topic in the RL community, many members of the ICLR community might wonder “what kind of interference”? A half-sentence like “where updates for some inputs degrade accuracy for others“ (copied from the paper’s introduction)  could suffice here.

In section 5.1, the authors have lines beginning with bolded “DQN”, “DQN-LTA”, “DQN-Large” et cetera. For readability’s sake, it might be useful to format these as bulleted lists.

A small grammatical issue: “This issue is usually called gradient vanish”. Maybe rephrase this as “This issue is known as the vanishing gradient problem”.

The authors list ReLUs as an example of an activation function with vanishing gradients. As vanishing gradient problems go, ReLU is a bit different from tanh and other nonlinearities that saturate, so I would avoid listing it here to avoid unnecessary debates.

The authors write “Mnist” in a number of places. The correct spelling is “MNIST”: this is an acronym for “Modified National Institute of Standards and Technology”.

---

> ### Author Response · Authors · 2020-11-21
> **Response to R3**
>
> We thank you for reading our paper and providing helpful feedback. We will incorporate the suggestions you have listed.
>
> We completely agree with the point that we had missing details for reproducibility. We have now added those details, in the updated paper. We have also given the paper to a colleague, to ask them to identify any missing details, and will incorporate those into a final paper. For some of the details you requested for the RL experiments, they had already been included in Section 5.1 and Appendix A.4 (reproducible research). We include them here to directly answer your question. For DQN, we use 64-by-64 (two-hidden-layer) ReLu. For DQN-LTA, we use 64-by-64 with the second layer using LTA, and hence, the output layer has (64*20=1280) * Number-of-Actions training parameters. For a fair comparison, we also use DQN-Large/L1/L2 as competitors. Those increase the number of hidden units in the second hidden layer to 1280 units so the weight matrix dimension in the output layer matches that of DQN-LTA.
>
> The primary concern is that we did not compare to state-of-the-art (SOTA) systems. As you point out, we could incorporate a new activation into many agents, so one could ask why not the best agents (as you are asking). The main reason is that our goal is to understand this new activation, rather than to get a SOTA system. SOTA requires more than just one new idea: it requires lots of engineering effort, implementation tricks, even specialized use of hardware. Further, such systems can be expensive to run, prohibiting more careful investigation with many runs and looking at more specific questions (like using sparsity regularizers, or looking at performance for varying delta and eta).
>
> We would further argue that DQN and DDPG are reasonable baselines in these benchmarking environments. It is meaningful to understand: how does incorporating this new activation impact performance, across multiple benchmark environments with two different types of algorithms (Q-learning and Policy Gradient)? Studying the impact across these settings is no doubt limited, and we cannot say for sure that the results will extend to say Atari with Rainbow. But, even if we saw improvements in Rainbow, this would not necessarily guarantee that we would then see those improvements extend to the real-world. The results we have, where we carefully sweep hyperparameters of other methods and include many runs to ensure results are less likely due to chance, provide some evidence that LTA does in fact improve robustness of learning, especially since the phenomena persists across a variety of benchmark problems with different agents.
>
> Please let us know if you need any further clarification.

---

### Official Review · AnonReviewer4 · 2020-10-29
**Interesting idea with convincing results, some issues in the sparsity claims though.**

**Rating:** 7
**Confidence:** 3

**Review:**

Summary:

This paper presents a new activation function "Leaky Tiling Activation", designed with the goal of learning sparse representations (where only a few units activate for a given input). This new activation function, rather than mapping a scalar to another scalar, maps a scalar to a smoothed one-hot representation based on bins., which encourages sparse representations.


Reasons for score:

Although I have some issues with the sparsity claims of this paper (see my comments below), I found the paper intriguing and I am curious to try similar ideas in other problems. So, with some work on the sparsity argument sections, I think this could be an interesting paper with ideas people would want to play with.


Additional feedback:

- page 3: "In the following theorem" -> which theorem? [edit after reading further: Do you mean Theorem 1 in Section 3.3? (probably not, as that's for LTA, not TA)]
- I am not sure if "leaky" is the best name for LTA, as more than "leaky", it is almost like the operators in fuzzy logic, with the term "leaky", I was imagining it more like a Leaky ReLU, where derivative is non zero everywhere.
- page 4: about the "guaranteed sparsity". I do not think it's very fair to compare the sparsity achieved by this operation with that achieved via regularization for the following reason: LTA increases the number of outputs of the unit. So, if in a regular layer with, say, a ReLU activation, you would have n outputs, with an LTA activation you would have k*n outputs. Assuming TA, instead of LTA, where will be exactly "n" outputs that are non zero, the same as in the original network with ReLU. So, LTA is not increasing sparsity, but just encoding the output in a different way that makes it look sparse as it increases the number of outputs (Theorem 1 and its Corollary feel totally unnecessary, as the network is sparse by construction, no need to prove it). It's like claiming that a OHE representation of a discrete label is "sparser" than the label itself, which is an iffy argument, as they encode the same information. I am not implying that LTA is not useful (I still find the paper very intriguing!), but just that the sparsity argument is questionable, and in my view it is not fair.
- page 5: it seems all results are in the appendix, which is unfortunate, as papers should stand on their own and appendix only used for additional explanation, not for main results. So, I am going to judge the paper just based on what is on the first 10 pages, to be fair to other papers. I'd expect sentences like "as figure X shows, blah, blah (more details in appendix)", but not "as figure X in the appendix shows, blah blah main result". This is trying to game the page limitations.

---

> ### Author Response · Authors · 2020-11-21
> **Response to R4**
>
> We thank you for reading our paper and providing constructive and helpful feedback.
>
> 1. “The following theorem 1 …” Thanks for pointing this out. We had moved that result to the appendix, and called it a Proposition, and did not appropriately point to it in the main body. It is updated in the current version.
>
> 2. We used the term Leaky because we modified flat regions to be sloped so the gradient information can be leaked into previous layers. But your suggestion is a good one. We will consider if we should change the name to Fuzzy Tiling Activation.
>
> 3. There was a small misunderstanding here, due to the fact that we did not clearly state the architecture for DQN-L1/L2. These methods had exactly the same size output layer as DQN-LTA (in your example, they had kn nodes in the second hidden layer). For these methods we took DQN-Large and incorporated the regularizers on that final layer. This means both methods had the opportunity to take the more dense, smaller input layer (of size n) and project it to a higher-dimensional (and sparse) layer of size kn.
>
> As an additional note, though we absolutely agree that Theorem 1 and Corollary 1 are simple proofs, they help quantify levels of sparsity for users. Corollary 1 particularly suggests how to set parameters delta and eta to obtain desired sparsity levels. Such choices are not obvious simply from look at the LTA, and so merit an explicit result for clarity. The results are also not obvious from inspection, so simply stating them as fact without justification would be incomplete. We are not arguing here that the complexity of our theory is a selling point of this work, but rather want to clearly characterize this new activation.
>
> Please let us know if you still have concerns about the comparison. We are happy to provide any further clarifications/explanations.

---

### Decision · Program_Chairs · 2021-01-07
**Final Decision**

**Decision:**

Accept (Poster)

**Comment:**

This paper proposes a new sparsity-inducing activation function, and demonstrates its benefits on continual learning and reinforcement learning tasks.

After the discussion period, all reviewers agree that this is a solid paper, and so do I. I am thus recommending it for acceptance as a poster. Hopefully, such visibility (combined with the open source release of the code) will encourage other researchers to try this new technique, and we will see more evidence confirming its usefulness in more varied settings and versus stronger baselines (that remain somewhat limited in the current work: this is the main weakness of the paper).